# Solving the Optimal Selection of Wellness Tourist Attractions and Destinations in the GMS Using the AMIS Algorithm

Rapeepan Pitakaso [1], Natthapong Nanthasamroeng [2], Sairoong Dinkoksung [3], Kantimarn Chindaprasert [4], Worapot Sirirak [5], Thanatkij Srichok [1], Surajet Khonjun [1], Sarinya Sirisan [6], Ganokgarn Jirasirilerd [6] and Chaiya Chomchalao [7],*

1 Artificial Intelligence Optimization SMART Laboratory, Industrial Engineering Department, Faculty of Engineering, Ubon Ratchathani University, Ubon Ratchathani 34190, Thailand
2 Artificial Intelligence Optimization SMART Laboratory, Engineering Technology Department, Faculty of Industrial Technology, Ubon Ratchathani Rajabhat University, Ubon Ratchathani 34000, Thailand
3 Ubon Ratchathani Business School, Ubon Ratchathani University, Ubon Ratchathani 34190, Thailand
4 Faculty of Tourism and Hotel Management, Mahasarakham University, Maha Sarakham 44000, Thailand
5 Department of Industrial Engineering, Faculty of Engineering, Rajamangala University of Technology Lanna Chiang Rai, Chiang Rai 57120, Thailand
6 Department of Industrial Management Technology, Faculty of Liberal Arts and Sciences, Sisaket Rajabhat University, Sisaket 33000, Thailand
7 Department of Industrial Technology, Faculty of Industrial Technology, Nakhon Phanom University, Nakhon Phanom 48000, Thailand
* Correspondence: chaiya_npu@ubu.ac.th

**Abstract:** This study aims to select the ideal mixture of small and medium-sized destinations and attractions in Thailand's Ubon Ratchathani Province in order to find potential wellness destinations and attractions. In the study region, 46 attractions and destinations were developed as the service sectors for wellness tourism using the designed wellness framework and the quality level of the attractions and destinations available on social media. Distinct types of tourists, each with a different age and gender, comprise a single wellness tourist group. Due to them, even with identical attractions and sites, every traveler has a different preference. A difficult task for travel agencies is putting together combinations of attractions and places for each tourist group. In this paper, the mathematical formulation of the suggested problem is described, and the optimal solution is achieved using Lingo v.16. Unfortunately, the large size of test instances cannot be solved with Lingo v16. However, the large-scale problem, particularly the case study in the target area, has been solved using a metaheuristic method called AMIS. According to the computation in the final experiment, AMIS can raise the solution quality across all test instances by an average of 3.83 to 8.17 percent. Therefore, it can be concluded that AMIS outperformed all other strategies in discovering the ideal solution. AMIS, GA and DE may lead visitors to attractions that generate 29.76%, 29.58% and 32.20%, respectively, more revenue than they do now while keeping the same degree of preference when the number of visitors doubles. The attractions' and destinations' utilization has increased by 175.2 percent over the current situation. This suggests that small and medium-sized enterprises have a significantly higher chance of flourishing in the market.

**Keywords:** wellness tourism; optimal selection of tourist attractions; family wellness tourists; group preferences; AMIS; mixed integer programming

## 1. Introduction

Wellness tourism is the practice of traveling in an effort to maintain or enhance one's own well-being. The desire for authentic experiences, a healthy lifestyle, avoiding disease, managing stress and limiting unhealthy lifestyle choices are its motivating factors. With the aid of the wellness industry, people may reclaim travel as a method of rest, relaxation,

renewal, adventure, joy and self-actualization. The wellness travel market, which was forecast to be worth 801.6 billion USD in 2020, is anticipated to expand by 7.2 percent annually from 2021 to 2030 to reach 1592.6 billion USD. People routinely visit hospitals, clinics, wellness spas, fitness facilities and wellness resorts in order to improve their physical and emotional health. Due to this shift in perspective, wellness tourism—which emphasizes enhancing one's health and sense of well-being through diverse physical, psychological and spiritual activities—has emerged. Additionally, it offers a range of services, such as accommodations, dining, shopping and others. Tourists desire to interact with people and discover their nature and culture in order to live a healthy lifestyle, lower stress, prevent disease and increase their well-being. As a result, there are more of them, which is expected to enhance demand for the wellness tourism industry.

Thailand is a fascinating country because of its exotic landscape, well-known Thai food, affluent metropolises, amazing attractions, five-star hotel and hospital service standards, wonderful health spas and wellness retreats. Thailand's people, who are friendly and welcoming whether they reside in a tranquil mountain village or the busy metropolis center of Bangkok, are what genuinely set the country apart. As a result, the wellness tourism industry in Thailand is growing. The Great Mekong Sub-region countries are centered around Ubon Ratchathani (GMS). Figure 1 shows the GMS transportation network, with a focus on Ubon Ratchathani Thailand. With over 10 borders and more than six of the continents' key roads (corridors), it can transport travelers from China, Vietnam, Laos, Myanmar and Cambodia (corridors). A total of 1645.546 million people live in these countries, or 21.22 percent of the world's population.

The Mun River flows through the center of the plateaus and mountain ranges in the province of Ubon Ratchathani. The "Emerald Triangle" refers to the region where Ubon Ratchathani borders both Cambodia and Laos because of its stunningly lush surroundings. Two of Isan's most pristine and unexplored natural reserves, Phu Chong Nayoi and Pha Taem national parks, add to Ubon Ratchathani's natural attraction. The annual candle festival, a beautiful Buddhist celebration, is a fantastic time to visit Ubon Ratchathani, the region's major city. One can visit Pha Taem National Park, known for its ancient rock art, Sam Pan Boak, also known as the Grand Canyon of Thailand, Huai Sai Yai Waterfall, one of the most stunning waterfalls in Northeastern Thailand, Kaeng Saphuee Public Park, Wat Tham Khuha Sawan and Wat Phrathat Nong Bua, among many other fascinating and lovely locations in Ubon Ratchathani. The Khao Phansa Day Candle Festival, in which enormous, intricately carved candles are paraded across the city, is the most significant event of the province for festival enthusiasts.

Ubon Ratchathani is genuinely poised to be a tranquil city that is the heart of wellness tourism in Northeastern Thailand, attributed to the variety of nature, atmosphere, shopping, symbolic landscape, culture and archaeological sites. As we have already mentioned, wellness tourism refers to vacation activities that tourists engage in to enhance both their physical and emotional well-being. The government promotes Phuket, Krabi, Bangkok and Chiangmai as Thailand's top wellness tourist destinations, as is claimed in [1]. Ubon Ratchathani is one of Thailand's most fascinating wellness tourism destinations, and as a result, it is made up of the locations and attractions that serve as the main hubs for the wellness industry's operations.

Because no single location can provide the whole range of necessary wellness packages, Ubon Ratchathani is unprepared for wellness tourism. Ubon Ratchathani, however, has the capacity to meet the demands of high standards in wellness tourism. In this study, we create a model for delivering wellness services from a variety of Ubon Ratchathani service providers in high-quality, comparable wellness packages to visitors for the GMS region.

In [2], the existence of several wellness-related attractions/destinations (WA/D) is demonstrated. The WA/Ds include things such as vacation spots in the wild and in nature, spa packages, yoga, meditation, healthy eating and fitness, multigenerational raves, self-improvement programs, cooking classes, art workshops, nutritional counseling and stress-relieving activities. All of these wellness vacation spots and attractions may be

divided into four categories: (1) stress relief (SR), (2) physical health improvement (PI), (3) mental health improvement (MI) and (4) sickness prevention (A/D) (SP). Using this concept, we create high-quality WA/D from Ubon Ratchathani's small and medium-sized wellness attractions and destinations. Ubon Ratchathani must be able to compete with other popular wellness resorts found in other regions of Thailand.

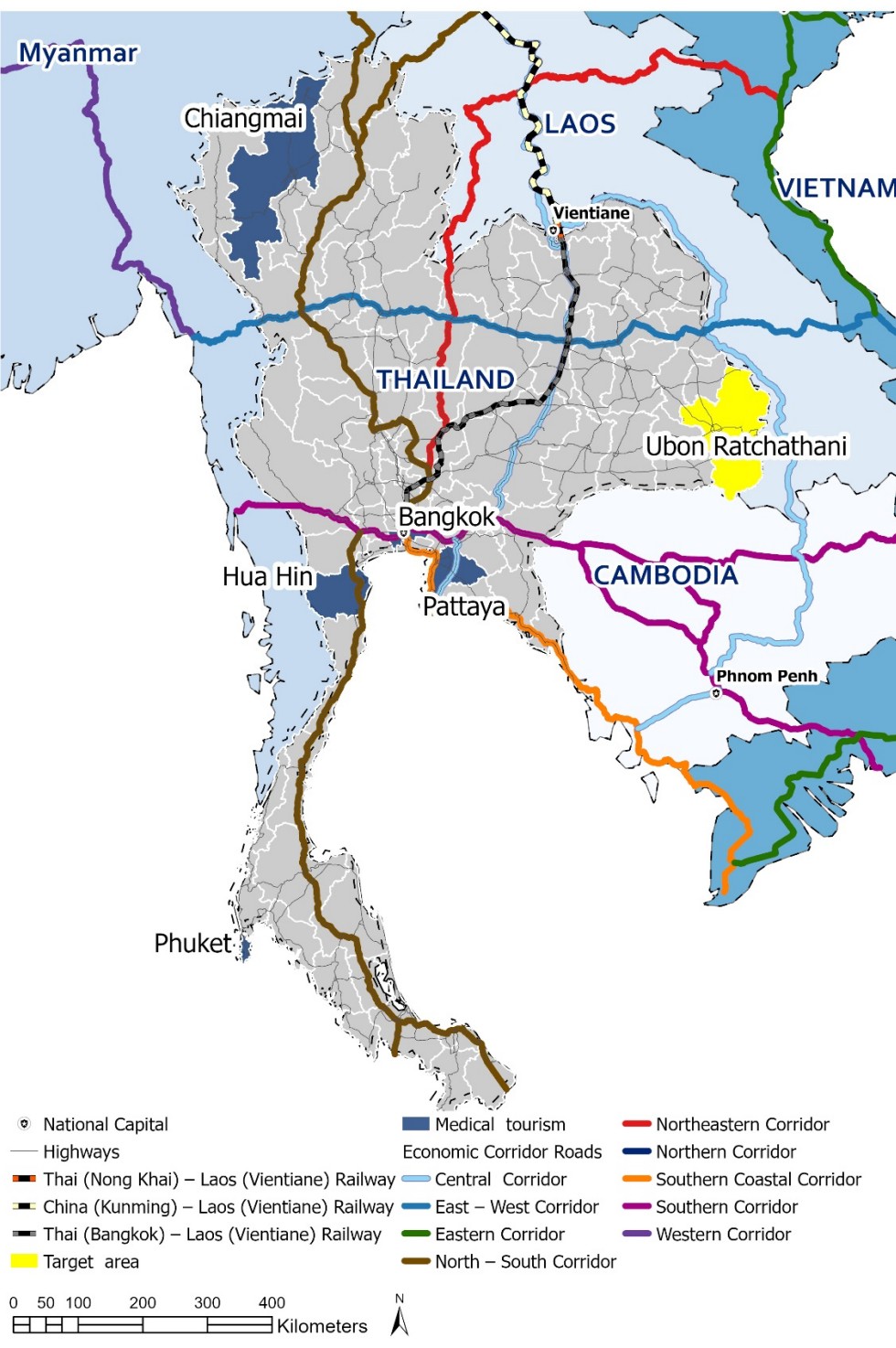

**Figure 1.** Main roads and borders of GMS region. (source: Environment Operation Center, www. gms-eoc.org, access date 20 July 2022).

All of the tourist attractions in Ubon Ratchathani are filtered out using Figure 2 to leave just the wellness-like attractions. To accommodate wellness travelers, the remaining attractions are then divided into four groups, each of which is combined to create a collection of attractions that are appropriate for the group of wellness tourists and that are part of the wellness package. The satisfaction of the travelers guides the creation of the vacation packages. According to [3], wellness tourists typically travel as family members or as part of large groups on tours such as retirement vacations. Members of a group tour or family vacation are typically of varying ages and genders. The authors of [4] make the point that people of different ages and genders have diverse preferences for places and attractions, making it more difficult to choose these items for a given tourist group. When establishing the tourist's preference for the attractions, the age and gender of each target group member are taken into consideration. The preferences of the tourist are ascertained using the survey from [4].

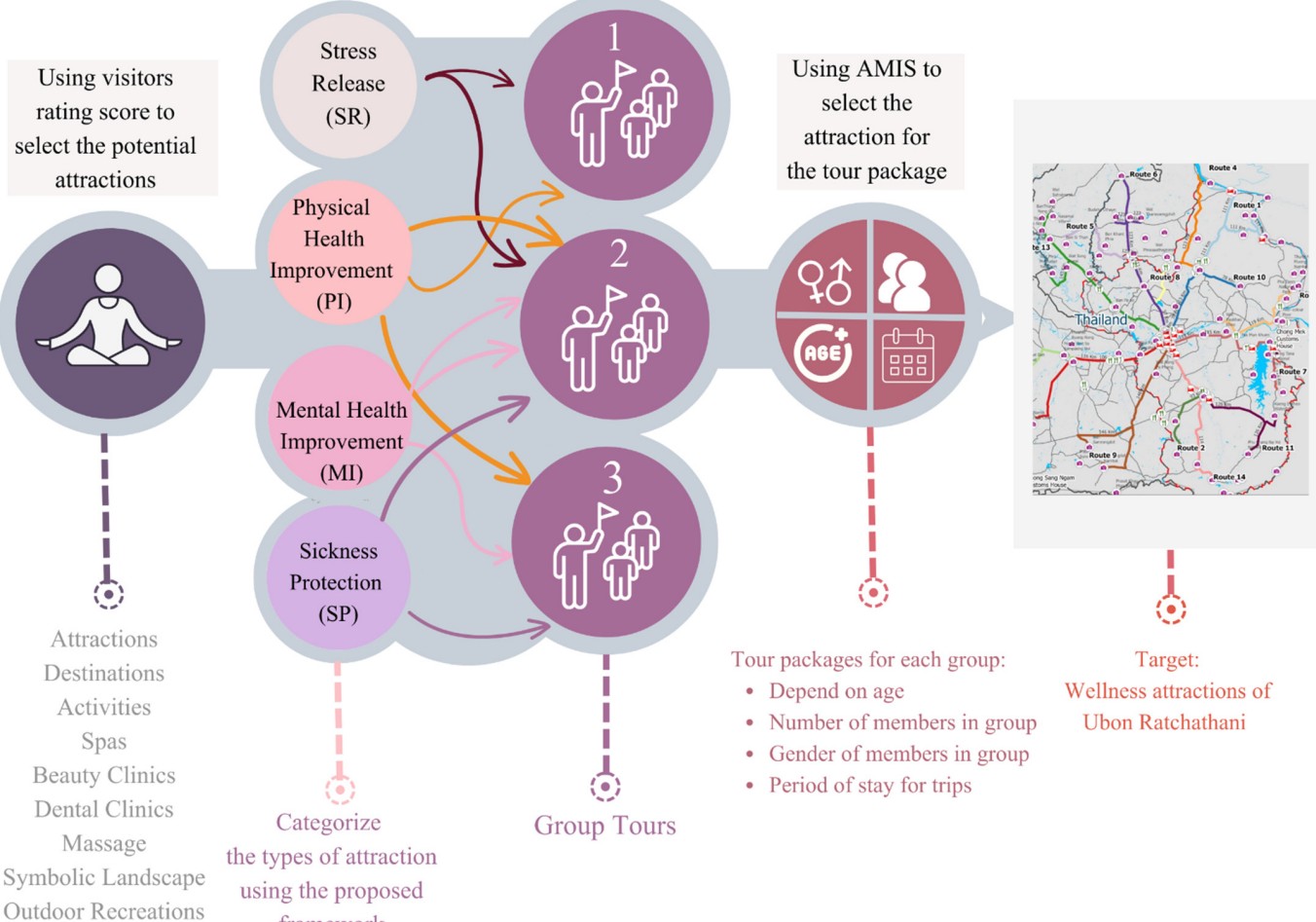

**Figure 2.** Framework of the proposed.

The following are some ways that this paper benefits both the academic and business worlds. The first two contributions are discussed as follows, with a novel mathematical model and a useful approach to solving it being suggested. The suggested strategy can be used by researchers to enhance their methods as they attempt to improve solution approaches for various types of problems. The latter three terms of contributions deal with social and business implications. The model suggested in this study can be used by business sectors to increase the competitiveness of their entire supply chain firms.

(1) A mathematical model formulation is developed to select the finest destinations and attractions to combine with other locations to build the most practical wellness vacation package for the GMS populations.

(2) An effective approach is proposed and evaluated with well-known heuristics proposed in the literature in order to solve the mathematical model presented in (1).

(3) Based on the preferences of the family or tourist group, the optimal destinations to visit are selected.

(4) A new management system for the SME wellness organization is made available as the collaborative business partners in the wellness sectors.

(5) To determine the impact of altering the system's capacity for tourists, a sensitivity analysis of the case study is presented. As a result, the hotel, tourist attractions and restaurants may get ready for the incoming travelers.

The rest of the text is organized as follows. In Section 2, the literature review and relevant investigations are presented. In Sections 3 and 4, the problem statement and the formulation of the mathematical model and proposed algorithms are presented. Sections 5 and 6, respectively, present the computational results and conclusion.

## 2. Literature Review and Related Work

The word "wellness" has a long history. Wellness has grown in prominence as a contemporary idea since the 1950s when the writings and leadership of a loosely connected group of American doctors and philosophers significantly influenced how we perceive and talk about well-being today. However, wellness has much earlier origins—possibly even prehistoric ones. Many of the intellectual, theological and medical revolutions that occurred in Europe and the United States in the 19th century have influenced various aspects of the wellness concept. The prehistoric communities of Asia, Greece and Rome are also where the concepts of well-being can be discovered. These societies' historical traditions have had a significant influence on the modern wellness movement [5]. Regarding health between 3500 and 1500 BC, the Vedas, a group of four sacred Hindu texts, recorded the early oral tradition of Ayurveda. Ayurvedic regimens are tailored to each person's distinctive constitution (their dietary, physical activity, social contact and cleanliness requirements) with the goal of preserving a balance that wards off sickness. An extensive philosophy known as Ayurveda aims to encourage balance between the body, mind and spirit. Yoga and meditation are fundamental to the tradition in addition to becoming more and more well liked on a global scale. Traditional Chinese medicine has undergone continuous development between 3000 and 2000 BC in response to Taoism and Buddhism, the Ancient Greek physician Hippocrates and the focus on disease prevention in Ancient Roman medicine.

In [6], wellness tourism is defined as the practice of traveling for the purpose of receiving beneficial therapies that help to preserve health and enjoy life. Life quality and wellness are related. With the help of treatments and therapies that take a holistic approach to health, the vital equilibrium and balance between the body, mind and spirit can be restored. The result of this harmony, which both heals and rebalances the energy flow, is total health. Therefore, all excursions and activities that meet these requirements are referred to as wellness tourism. "Tourist well-being" is one of the most frequently used phrases to describe the wellness traveler. Tourist well-being can be defined as the subjective pleasure and sense of personal development that travelers feel after completing their vacation goals and satisfying various sensory demands [7,8]. Based on a psychological study on well-being [9–12], it is increasingly believed that tourist well-being has two dimensions: eudaimonia and hedonia. Tourist eudaimonia depicts travelers' inner sentiments of self-growth, such as achieving individual potential and self-fulfillment, whereas tourist hedonia shows visitors' emotional pleasure from satisfying their sensory expectations, such as enjoying delicious meals or magnificent scenery [13]. Contrary to tourist hedonia, which may reflect the concrete aspect of a visitor's well-being because it is concerned with sensory pleasure, a lower-order fundamental human need, tourist eudaimonia may reflect

the abstract aspect of a visitor's well-being because it is concerned with realizing one's potential and self-fulfillment

Numerous academic articles have recently studied the connection between tourist-related activities and tourism well-being [14,15]. It is suggested that, in addition to basic travel requirements, tourists also have health-related needs. Nevertheless, wellness travel is becoming more and more popular and is important to the economics of the sector. Figure 3 depicts the outline of the wellness-tourism-related activities that we have deduced from the relevant literature.

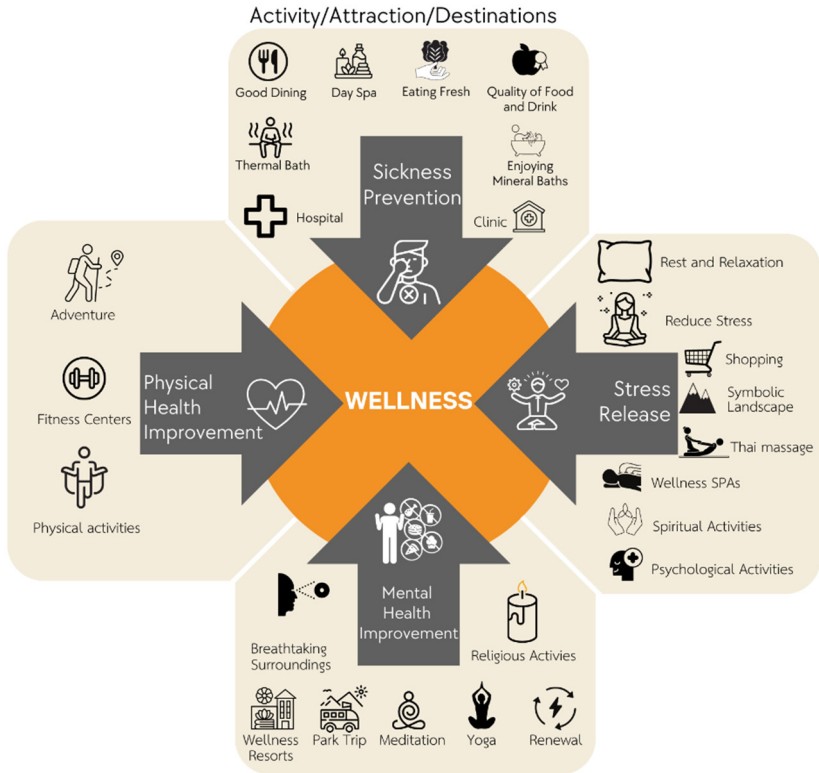

**Figure 3.** Wellness connection of aims of travelling and activities of wellness tourism.

Figure 3 shows that the four goals of wellness travelers are to (1) sickness prevention, (2) manage stress, (3) improve mental health and, (4) improve physical health. There are numerous sights and activities that can help travelers achieve their goals. Outdoor leisure, shopping, symbolic landscapes, wellness spas and other activities are among them. Some tourist activities may help them achieve multiple goals. Today's wellness tourists look for more than just a wonderful hotel or resort; they also look for things to do while they are there that are listed in the wellness report. According to [16], the idea of well-being encompasses pleasure, health and spirituality in a more holistic way. Despite not typically being thought of as wellness tourism, outdoor and adventure recreation and travel can significantly improve health and well-being. In [17], the UNESCO forests, volcanic lava tubes and ocean are among the top attractions for wellness tourists that travel to Jeju Island. Due to Jeju's rich cultural and natural resources, which link to an all-encompassing place-based experience that safeguards and supports a peaceful and contented life, [18] addresses the fact that, although wellness travelers are still seeking out trip packages that include things such as yoga, meditation, nutrition and fitness, they are also increasingly attracted to wild and natural locales, and they additionally support the growing interest in intergenerational travel among wellness tourists. This particular visitor looks for vacation destinations that welcome families. The parcels should contain something for every member of the household. Golf and kayaking are wonderful for

parents, and ziplines and meandering rivers are great for kids. All of these characteristics work well together.

The activities associated with wellness tourism are detailed in [19]. It has been demonstrated that not only top-notch resorts and hotels must be ready for wellness tourists. The wellness traveler anticipates participating in certain events, sights and locations to improve the quality of their wellness trip. These include things such as thermal baths, yoga retreats, lifestyle retreats, wellness centers, day spas, salons and beauty parlors, wellness cruises and exercise facilities. The authors of [20] examine the effects of visits to outdoor adventure parks on day visitors' health. The results show that recreational activities in outdoor adventure parks have immediate effects on well-being that are also influenced by gender norms. The authors of [21] provide an illustration of the strong link between Chinese wellness tourism and longevity. Longevity reflects the holistic belief that place and health interact as a connected system. In the Chinese setting, the symbolic landscape, which is heavily affected by the longevity culture, plays a significant role in the healing process of the visitors, even though the natural environment, social interaction and symbolic landscape all play a part.

Many individuals' mental health has been significantly impacted by the COVID-19 pandemic, as is claimed in [22]. Exercise, reading and listening to music are just a few of the things that people should partake in to help maintain and enhance their mental health. According to [23], participating in adventure tourism activities may help elders overcome psychological obstacles even if they simultaneously experience physical ones. The links between tourism and general well-being are highlighted in [24]. They show that a person's level of well-being is a dynamic assemblage that depends on the performative impacts of their surroundings, geography, age, time and life experiences. Depending on individuals' early exposure to their native land and culture, the age of their first and subsequent returns, changes in their life goals and notions of well-being, four patterns of how diaspora tourism effects alter throughout the course of a person's lifetime have been discovered. Based on their own expressions of experiences, meanings, emotions and life goals upon returning home, the traveler likes these patterns.

Travelers expect wellness tourism to include a range of health services, such as traditional and alternative medical care, as well as tourist attractions, in order to improve their health and well-being (HWB), according to research [19]. An advantageous trade-off between expectations and perceptions of wellness tourism has a good effect on tourist HWB and behavioral intention. Finding a good resort or hotel and engaging in healthy activities such as hiking, sporting excursions, biking, sightseeing in beautiful settings, indulging in delectable cuisine, unwinding in a welcoming environment, etc., are the new trends for wellness tourists, according to the related literature. In this study, the ideal attractions and locations are combined with the available wellness vacation packages. All of the attractions in the target area are eliminated to leave only the ones that are appropriate for wellness tourists, and the packages are then created to have the lowest cost for the travel agency while maintaining the best standards for group tours.

Families and friends comprise the majority of wellness tourists. According to [25], the majority of wellness visitors are young people. These healthy travelers learn about their travel plans from friends and internet news sources. Except for Japanese guests, who are largely elderly and stay in condominiums for their extended vacations, everyone else travels with family and friends and stays at hotels and resorts. The majority of visitors go on vacation for their personal pleasure in order to take in the gorgeous surroundings as well as to unwind and revitalize. Travelers are drawn to experiences such as Thai massages, eating wholesome regional cuisine, visiting spas and taking mineral baths. Safety and the standards of food and drinks are prioritized by the majority of wellness travelers. According to this study, most people who travel for wellness do so with friends and family. Different tourist destinations or attractions appeal to visitors of different sexes and age groups. According to [23,26,27], different human traits, such as age, gender, etc., have varying requirements in the attractions or destination. Examples include the fact that older

visitors' health can be improved by visiting historical places, the fact that young people are more awed by stunning landscapes, the fact that women prefer yoga to biking, etc.

In [28], enjoyable activities are created to prepare for family vacations. It requires an all-encompassing family strategy because educating children about these visitors is equally vital from both their parents' and children's viewpoints. On the other hand, kids have an impact on how a family vacation is seen; thus, companies who want to provide interesting, difficult and pleasurable family vacations that satisfy both parents and kids need to understand kids in order to do so. Rojas-de-Graci and Alarcón-Urbistond [29] assert that it is impossible to gauge how happy a family is with a trip without also taking the happiness of the other family members into account. The entire tourist group, including the parents, spouses and children, must be content. Each member's pleasure affects the other members. When participating in outdoor recreational activities with their families, adventure service providers should take each family member's level of adventure into account, according to [28]. As a result, both individual and group satisfaction can be maintained. Additional research demonstrates that people of different ages and genders have different needs and levels of enjoyment from the same attraction or activity [30–36].

The plan for creating the travel itinerary for family or group tours is laid out in [4]. A distinct tourist route, then an individual route, is formed as a result of the gender and age of the group tour participants by varying the requirements and level of enjoyment for the same place. The wellness tourism industry has grown tremendously, and its value now has an impact on important economic principles. The operational level of the attractions/destinations chosen to be included in the wellness packages is provided in this study in order to increase the competitiveness of wellness service providers. The authors of [37] address that collaboration within and across the businesses can be a key success element. Today, companies have established both official and informal network structures based on trust, where cooperation between them plays a vital role. Almost every industry today, regardless of size, can attest to the economic significance of supply chains. Big corporations and even small and medium-sized businesses are actively participating in global value-creating chains. Moreover, [38] presents 12 ways to compete with large businesses by using SMEs. Three keywords to success in this manner are to create local content, local promotion and local market concern. Therefore, from our study of all the reviewed articles, the selection of attractions/destinations to form quality wellness tourist packages is the business model for SME to collaborate and obtain compatible wellness packages with large wellness tourism providers. It is not only the negotiation that initiates the cooperative business, but it is also the appropriate business matching that enhances the province's overall image in the wellness tourism industry. The model incorporates selection, routing and attraction recommendations. Based on historical data, the package is created by combining the group tour preferences. The multi-objective attraction/destination selection for the wellness family package tour (M-ADes-SFWFPT) is presented as a mathematical model, and effective heuristics to solve the problem and to compare with well-known heuristics, such as the genetic algorithm, differential evolution algorithm and variable neighborhood strategy adaptive search, are presented.

Due to M-ADes-SFWFPT, multi-operation multi-machine scheduling is a particular example and is classified as an NP-hard issue [39]. However, in order to address the issue, efficient heuristics must be offered. We build the metaheuristics for this investigation using the AMIS given by Pitakaso et al. [40]. The difficulty of designing a multi-echelon logistics network has been effectively solved using AMIS. There are simply a few steps and inputs required. The most current metaheuristics to be proposed in recent years is the Artificial Multiple Intelligence System (AMIS). In the past, numerous metaheuristics have been employed to address millions of challenging issues. A well-known metaheuristic is the genetic algorithm (GA). It has been used to resolve a variety of issues, including timetabling, scheduling issues and other issues involving engineering optimization [41]. The differential evolution algorithm, known as Differential Evolution (DE), is a technique that seeks to enhance a candidate solution through time in order to maximize a problem's

potential. In the 1990s, Storn and Price introduced DE [42,43]. Numerous types of problems, including the three-dimensional packing issue [44], the closed loop facility layout issue [45] and flexible job shop scheduling with outsourcing operations [46], have been handled using it.

Other effective metaheuristics that are used to tackle state-of-the-art issues include Simulated Annealing [47], Tabu Search [48] and particle swam optimization (PSO) [49]. Since we researched the prior literature, we were unable to identify any strategy that could handle the stated problem in a direct manner. Although Pitakaso et al. [40] identify AMIS as a quick and efficient technique, we decided to modify it to address the stated challenge.

## 3. The Mathematical Model Formulation for M-ADes-SFWFPT

The mathematical model formulation is presented step by step in this section as follows. The objectives function of the M-ADes-SFWFPT is to minimize the total cost of the system and to maximize the total group preference of the tourists.

Indices

| | |
|---|---|
| $s$ | hotel s when $s = 1 \ldots S$ |
| $i$ | attractions/destinations $i = 1 \ldots I$ |
| $j$ | tourist group $j = 1 \ldots .J$ |
| $n$ | type of tourist $n = 1 \ldots N$ |
| $k$ | type of attraction/destination $k = 1 \ldots K$ |
| $t$ | planning period $t = 1 \ldots T$ |

Parameters

| | |
|---|---|
| $c_i$ | the cost incurred when attraction $i$ is included in any package (Baht/time/person) |
| $p_i$ | capacity of attractions/destinations $i$ for each time period (person) |
| $v_{njt}$ | the tourist type $n$ joins package $j$ period $t$ (person) |
| $o_i$ | a popularity score of attraction $i$ (score) |
| $u_{ni}$ | the preference score of tourist type $n$ for attraction $i$ (score) |
| $\varnothing_s$ | the hotel's per-person, per-night rate (Baht/night) |
| $\Omega$ | full preference score set to 10 |
| $w_{ik}$ | equals 1 if attraction $i$ is classified as type $k$ of wellness destinations and 0 otherwise |
| LN | large number which is defined as 500,000 |

Decision Variables

| | |
|---|---|
| $A^1_{sj}$ | $\begin{cases} 1 & \text{if hotel } s \text{ is assigned to } j \\ 0 & otherwise \end{cases}$ |
| $v^2_{jt}$ | $\begin{cases} 1 & \text{if it has a positive number of tourists in package } j \text{ period } t \\ 0 & otherwise \end{cases}$ |
| $g^1_{nj}$ | number of tourists of type $n$ participating in group tour $j$ |
| $d^1_{ij}$ | total preference and popularity score of attraction $i$ to group tour $j$ |
| $X^2_{ijt}$ | equals 1 and 0 the other way around |
| $d^2_{ij}$ | percentage of the tourist group $j's$ satisfaction with attraction $i$ |
| $d^3_{kj}$ | attractiveness of travel packages $j$ to type $k$ of attraction |
| $d^4_{kj}$ | equal to 1 when $d^3_{kj}$ is more than 20% and 0 otherwise |

Objectives Function

$$Min \ Z^1 = \sum_{n=1}^{N} \sum_{t=1}^{T} \sum_{j=1}^{j} \sum_{i=1}^{I} \frac{(u_{ni} + o_i)}{\Omega} c_i v_{njt} X^2_{ijt} + \sum_{n=1}^{N} \sum_{t=1}^{T} \sum_{j=1}^{J} \sum_{s}^{S} \varepsilon_j \varnothing_s A^1_{sj} v_{njt} \tag{1a}$$

And

$$Max \ Z^2 = \sum_{n=1}^{N} \sum_{t=1}^{T} \sum_{j=1}^{j} \sum_{i=1}^{I} (u_{ni} + o_i) v_{njt} X_{ijt} \tag{1b}$$

Subject to

$$\sum_{i=1}^{I} X_{ijt} \leq 1 \quad \forall j = 1\ldots, J, \ t = 1\ldots T \tag{2}$$

$$\sum_{n=1}^{N} \sum_{j=1}^{J} \sum_{n=1}^{N} v_{njt} X_{ijt} \leq p_i \quad \forall i = 1\ldots, I, \ t = 1\ldots T \tag{3}$$

$$\sum_{n=1}^{N} v_{njt} \leq LN v_{jt}^2 \quad \forall j = 1\ldots, I, \ t = 1\ldots T \tag{4}$$

$$\sum_{i=1}^{I} X_{ijt} = v_{jt}^2 \quad \forall j = 1\ldots, I, \ t = 1\ldots T \tag{5}$$

$$X_{ijt} \leq v_{jt}^2 \quad \forall j = 1\ldots, I, \ t = 1\ldots T, i = 1\ldots I \tag{6}$$

$$d_{ij}^1 = \sum_{n=1}^{N} g_{nj}^1 (u_{ni} + o_i) \quad \forall i = 1\ldots, I, \ j = 1\ldots J \tag{7}$$

$$g_{nj}^1 \geq v_{njt} \quad \forall n = 1\ldots, N, \ j = 1\ldots J \ and \ t = 1\ldots T \tag{8}$$

$$d_{ij}^2 = \frac{d_{ij}^1}{\sum_{i=1}^{I} d_{ij}^1} \times 100\% \tag{9}$$

$$d_{kj}^3 = \sum_{n=1}^{N} \sum_{i=1}^{I} (u_{ni} + o_i) w_{ik} g_{nj}^1 \quad \forall k = 1\ldots, K, \ j = 1\ldots J \tag{10}$$

$$d_{kj}^3 = \frac{d_{kj}^3}{\sum_{k=1}^{K} d_{kj}^3} \times 100\% \tag{11}$$

$$d_{kj}^4 = \begin{cases} 1 \ if \ d_{kj}^3 \geq 20\% \\ 0 \ otherwise \end{cases} \tag{12}$$

The objective function of the proposed problem is divided into two objectives; therefore, this problem is defined as a multi-objective problem. First, the objective is to minimize the total cost in the system, and the second objective is to maximize the satisfaction of the tourist. The cost objective (1a) is composed of two terms. The first term is the cost of the assignment of the attractions to group tour j. The seconds objective (1b) is to maximize the preference score of the total tourists that are assigned by the various attractions during their staying periods. It is specified that only one task (attractions/destinations) can be assigned to the package per time period. One time period can be defined as 3,4,5,6 h depending on the travel planner. In this case, one period equals 24 h. This constraint is depicted by Equation (2).

Constraint (3) ensures that, in each time period t, the capacity of attractions cannot be violated from servicing too many tourists. Constraint (4) defines the relationship of the binary variable $v_{jt}^2$ and the real number parameter $v_{njt}$, such that the binary is equal to 1 only when it has positive value of $v_{njt}$. LN is a large number, which can be defined as 500,000, and the relationship of $X_{ijt}$ and $v_{jt}^2$ is that it must have at least one attraction that is assigned to package j period t if $v_{jt}^2$ is equal to one (Constraint (5)). Constraint (6) indicates that $X_{ijt}$ cannot appear to equal 1 if there is no demand from the tourist in that period. Constraint (7) is used to generate the first method for obtaining the tourists' wish lists. Let $d_{ij}^1$ be defined as the preference score of the travel package $j$ to the attraction $i$. The number of participants in a package tour $j$ belonging to category $n$ of tourists is defined as $g_{nj}^1$, and $g_{nj}^1$ is derived from Constraint (8). $d_{ij}^2$ is the percentage of the tourist group $j's$ satisfaction with attraction $i$, and it may be calculated using Constraint (9). Constraint (10) is used to calculate the attractiveness of travel packages $j$ to type $k$ of attraction $(d_{kj}^3)$. Let $g_{nk}^2$ be the number of tourists of type $n$ who prefer type $k$ of attractions. $d_{kj}^3$ is the percentage of the tourist group $j's$ satisfaction with attractions of type $k$, and it may be calculated using Constraint (11). Constraint (12) is used to derive the value of $d_{kj}^4$.

## 4. Artificial Multiple Intelligence System (AMIS)

The AMIS is a brand-new heuristic that the authors of [40] have suggested. Utilizing multiple sorts of improvement techniques to solve a particular topic is the fundamental tenet of the AMIS. The flexibility and precise balancing of exploration and exploitation search is the AMIS's greatest strength. The four phases that constitute the AMIS are: (1) creating the initial set of work packages (WP); (2) carrying out the WP execution process; (3) updating the heuristics information; and (4) repeating steps (2) and (3) until the termination condition is satisfied.

Table 1 displays an illustration of the initial set of work packages. Each of the five task packages in Table 1 comprises ten entities (components), which correspond to the number of attractions and destinations.

**Table 1.** The initial set of work packages includes five WPs that are chosen at random.

| | Hotel | | | Attractions | | | | | | | | | |
|---|---|---|---|---|---|---|---|---|---|---|---|---|---|
| **WP** | **1** | **2** | **3** | **A** | **B** | **C** | **D** | **E** | **F** | **G** | **H** | **I** | **J** |
| 1 | 0.18 | 0.01 | 0.63 | 0.05 | 0.56 | 0.84 | 0.67 | 0.46 | 0.78 | 0.30 | 0.26 | 0.81 | 0.75 |
| 2 | 0.06 | 0.84 | 0.08 | 0.12 | 0.33 | 0.97 | 0.13 | 0.59 | 0.45 | 0.98 | 0.55 | 0.94 | 0.41 |
| 3 | 0.55 | 0.91 | 1.00 | 0.83 | 0.51 | 0.26 | 0.91 | 0.10 | 0.45 | 0.20 | 0.40 | 0.10 | 0.00 |
| 4 | 0.11 | 0.25 | 0.98 | 0.40 | 0.13 | 0.65 | 0.01 | 0.23 | 0.83 | 0.42 | 0.09 | 0.87 | 0.45 |
| 5 | 0.78 | 0.52 | 0.93 | 0.49 | 0.88 | 0.90 | 0.71 | 0.81 | 0.20 | 0.03 | 0.05 | 0.99 | 0.68 |

The work package is divided into two parts. The first part is the hotel selection, and the second part is the attractions/destinations selection. The first work package has 13 entities, whose values are 0.18, 0.01 and 0.63 (hotel selection) and 0.05, 0.56, 0.84, 0.67, 0.46, 0.78, 0.30, 0.26, 0.81 and 0.75 (attraction selection), respectively. The proposed problem is solved by decoding the WP, and the decoding process may be described step-by-step as follows.

**Decoding Method**

Step 1: Count all the tourists composed in each tourist group, and then sort the data into List A in ascending order. For instance, List A = 1, 4, 2, 5, 3.

Step 2: Sort the value of entities in the hotel, coded in an increasing order. This list is called List B, and WP 1 has List B = {2,1,3}. Then, we assign the hotel to the tourist group according to List A and List B. The assignment of the hotel must be under the hotel's capacity.

Step 3: After assigning the attractions/destinations to the group tours (package) in the first order of list A (in this case, group tour 1) and waiting for group 1 to complete all of its requirements, assign group 4, 2, 5 and 3 one at a time. The assignment of the attractions to the values of $d_{ij}^1$ and $d_{kj}^4$ is performed in the following way:

Step 3.1: Choose the tourist destinations first whose category has a value of $d_{kj}^4$ equal to 1. The attraction with the lowest value of WP is chosen first if there is more than one attraction to choose from using this criterion.

Step 3.2: Add the attraction with the highest value of $d_{ij}^1$ to the package once all types of $k$ with values of $d_{kj}^4$ equal to 1 have been chosen. If both candidates have equal values for $d_{ij}^1$, the candidate with the higher WP value is chosen first.

Step 3.3: Until all tourist stay times are satisfied, repeat process 3.1. Please ensure that the attractions' capacity limits are always followed.

Step 3.4: If *i* is assigned to *j* more than once to the attraction, transfer all *i* to be serviced in continuous periods. For example, if the outcome of package 1's assignment in periods 1 to 4 is "A,B,A,C," the new assignment is "A,A,B,C."

Step 4: Repeat step 2 until all packages and time frames are appropriate.

An example of using the decoding method for work package number 1 is shown in Table 2. The obtained result is that tourist groups 1,2,3,4 and 5 are serviced by 10 attractions, which have a total cost value of 75,247 Baht and a total preferable score value of 8724 scores.

**Table 2.** Decoding result of Example 1.

| Tourist Group | Type of Tourist (Person) ($v_{njt}$) | | | | | Duration of Stay (Period) | Entering Period (t) | Hotel | Period | | | | | | |
|---|---|---|---|---|---|---|---|---|---|---|---|---|---|---|---|
| | N1 | N2 | N3 | N4 | N5 | | | | 1 | 2 | 3 | 4 | 5 | 6 | 7 |
| 1 | 2 | 4 | 7 | 0 | 1 | 3 | 1 | 2 | A | A | C | | | | |
| 2 | 2 | 4 | 4 | 2 | 4 | 5 | 3 | 1 | G | D | H | J | B | | |
| 3 | 4 | 8 | 8 | 5 | 6 | 2 | 5 | 3 | B | E | | | | | |
| 4 | 0 | 5 | 5 | 2 | 2 | 7 | 3 | 2 | J | C | G | F | A | A | E |
| 5 | 0 | 4 | 7 | 4 | 7 | 4 | 5 | 1 | A | D | F | I | | | |

Steps (2), (3) and (4) of the AMIS are modified from Pitakaso et al. [40] and can be explained briefly as follows. The goal of the WP execution process is to increase the quality of the present set of solutions by utilizing various sorts of improvement techniques. The improvement techniques in the AMIS are known as intelligence boxes. Pitakaso et al. [40] recommend that the number of intelligence boxes be no less than eight. When there are multiple intelligence boxes to choose from, Equations (13) and (14) are used to select the best intelligence box for each WP.

$$S_{bt} = \frac{FN_{bt-1} + (1-F)A_{bt-1} + KI_{bt-1}}{\sum_{b=1}^{B} FN_{bt-1} + (1-F)A_{bt-1} + KI_{bt-1}} \tag{13}$$

$$P_{bt} = \begin{cases} P^{Max} \ if \ S_{bt} \geq P^{Max} \\ S_{bt} \quad if \ P^{Min} \leq \ S_{bt} \leq \ P^{Max} \\ P^{Min} \ if \ S_{bt} \leq P^{Min} \end{cases} \tag{14}$$

where $S_{bt}$ is the attractiveness to choose $IB_b$ in iteration *t*. $P_{bt}$ denotes the probability of selecting $IB_b$ in iteration *t*. $p^{Min}$ and $p^{Max}$ are the minimum and maximum probabilities of selecting an *IB* (predefined parameters). The minimum value of *P* is set to 0.2, and the maximum value is 0.8 (obtained from the preliminary experiment). $N_{bt-1}$ is the number of WPs in the previous iteration that chose an intelligence box $(IB)_b$.

$I_{bt-1}$ is a reward value that grows by 1 if $IB_b$ finds the optimal solution in the latest iteration but is set to 0 otherwise; $A_{bt-1}$ is the average objective value of all WPs that select $IB_b$ in the previous iterations; *IB* stands for the total number of *IBs*; *F* stands for scaling (*F* = 2); and *K* stands for improvement (*K* = 1).

The chosen IB is executed iteratively by the WP. Nine intelligence boxes are employed in our research. Enhancing search capabilities is the key design principle behind the intelligence boxes. There are three different kinds of intelligence boxes that are used: the diversification search intelligence boxes (D_i), the intensification search intelligence boxes (I_n) and the search that combines the D_i and I_n types. In Table 3 and Equations (15)–(23), the intelligence boxes employed in this study are displayed.

**Table 3.** Intelligence box details [40].

| IB Operators | Group | Value of $Y_{ijq}$ | |
|---|---|---|---|
| ACO-inspired move (AIM) | $I_n$ | $Y_{ijq} = \rho Y_{rjq} + F1\left(B_j^{gbest} - Y_{rjt}\right) + F2\left(Y_{mjt} - Y_{rjt}\right)$ | (15) |
| PSO-inspired move (PIM) | $I_n$ | $Y_{ijq} = Y_{rjq} + F1\left(B_j^{gbest} - Y_{rjt}\right) + F2\left(B_{hj}^{pbest} - Y_{rjt}\right)$ | (16) |
| DE-inspired Move (DIM) | $I_n$ | $Y_{ijq} = Y_{rjq} + F1\left(Y_{mjt} - Y_{njt}\right)$ | (17) |
| ABCO-inspired move (BIM) | $I_n$ | $Y_{ijq} = Y_{rjq} + \varnothing_{rj}\left(Y_{rjt} - Y_{njt}\right)$ | (18) |
| Restart | $D_i$ | $Y_{ijq} = \mathbb{R}_{ij}$ | (19) |
| Random Transit (RT) | $D_i$ | $Y_{ijq} = \begin{cases} Y_{ijq-1} & if \ \mathbb{R}_{ij} \le CR^h \\ R_{ijq} & otherwise \end{cases}$ | (20) |
| Inter-Transit (IT) | $D_i$ | $Y_{ijq} = \begin{cases} Y_{ijq-1} & if \ \mathbb{R}_{ij} \le CR^h \\ Y_{njq} & otherwise \end{cases}$ | (21) |
| Scaling Factor (SF) | $D_i$ | $Y_{ijq} = \begin{cases} Y_{ijq-1} & if \ \mathbb{R}_{ij} \le CR^h \\ \mathbb{R}_{ij} Y_{ijq-1} & otherwise \end{cases}$ | (22) |
| RT-AIM | $I_n / D_i$ | $Y_{ijq} = \begin{cases} \rho Y_{rjq} + F1\left(B_j^{gbest} - Y_{rjt}\right) + F2\left(Y_{mjt} - Y_{rjt}\right) \ if \ \mathbb{R}_{ij} \le CR^h \\ Y_{ijq-1} \quad otherwise \end{cases}$ | (23) |

where $\mathbb{R}_{ij}$ is a random real number with a range of [0, 1], and $\varnothing_{rj}$ is a random real number running in the range of [−1, 1]. $B_j^{gbest}$ is the best WP created so far. F1 and F2 are scaling factors which, in this study, are chosen to be 0.5 and 0.5 as F1 and F2. When WP z decides to use IB b as the improvement method, z is categorized as the set Z, whereas other WPs that do not select IB b are categorized as set A. However, I = Z ∪ A when I is the total number of WPs.

$Y_{njt}$ and $Y_{mjt} A$ and $Y_{rjt} \in Z$. WP r is picked at random from set Z, and WP n and m are randomly selected from WP in set A. The position value evaporation rate, or $\rho$, is predefined as 0.05. Equation (24) is used to update the sub-iteration position of $Y_{ijq+1}$.

$$Y_{ijq+1} \begin{cases} Y_{ijq} & if \ f_{ir} \le f_{iq} \ and \ update \ \ f_{ir} = f_{iq} \ and \ Y_{rjq} = Y_{ijq} \\ Y_{rjq} & otherwise \end{cases} \quad (24)$$

The IB execution procedure is utilized to raise the caliber of the solutions. Each IB carries out $q$ iterations, where $q$ is a fixed, predetermined number. We refer to $q$ as a sub-iteration. The value of CR is set to 0.8 [40]. The objective function of WP in sub-iteration h is calculated using Equation (25).

$$f_{iq} = \sum_{v=1}^{V} w_v f_{iq}^v \quad (25)$$

where $f_{iq}$ and $f_{iq+1}$ are, respectively, the objective functions of $Y_{ijq}$ and $Y_{ijq+1}$. V is the number of objectives in the model, and $f_{iq}^v$ is an objective function for track $Y_{ijq}$ with $v$ = 1,2, 3, . . . $V$. It indicates that $w_v$ is the objective v's weight, and that $\sum_{v=1}^{V} w_v = 1$. There are two objectives in this scenario; $w_1$ is the weight of objective 1, which is chosen at random from U [0.1, 0.9], and $w_2$ is $(1 - w_1)$.

The default parameter $q$ is 100 iterations long for each IB [40]. To maintain the non-dominated solution, the Pareto front is utilized. Let $f^1(y_r)$ and $f^2(y_r)$ be the objective functions of tracks r's objectives 1 and 2, respectively. If we consider ° to be a collection of feasible solutions, then $y$ prevails over $y'$ if and only if $f^v(y) \le f^v(y')$ for all $v$ = 1,2,3, . . . ,$V$, $y$ is the set of decision vectors, and $f^v(y)$ is the set of objective functions. The analysis of the Pareto front's promising solution used in this article employs the Technique for the Order of Preference by Similarity to the Ideal Solution (TOPSIS), proposed Hwang and Yoon [50].

### 4.1. Updating the Heuristics Info

To use as the information for the following iterations, some heuristics data need to be modified. Table 4 displays the rule for the update.

**Table 4.** Updated role of the heuristics information.

| Variables | Updated Method |
|---|---|
| $N_{bt}$ | Total number of WPs from iteration 1 to iteration $t$ that choose IB $b$ |
| $A_{bt}$ | The overall IB that choose the IB's average objective value ( $\frac{\sum_{i=1}^{N_{bt}} f_{it}}{N_{bt}}$ ) |
| $I_{bt}$ | $I_{bt} = I_{bt-1} + G$ <br> when <br> $G = \begin{cases} 1 & \text{if the global best solution in iteration } t \text{ is included in black box } b \\ 0 & \text{otherwise} \end{cases}$ |
| $B_j^{gbest}$ | Current best global WP is updated |
| $B_{hj}^{pbest}$ | Updated IB's best WP is updated. |
| $R_{ijq}$ | Choose at random a value for all positions for each WP |

*4.2. Continue Carrying out the Work Package until All of the Termination Conditions Are Satisfied*

The stopping criterion in this context is either the predetermined computational time or the maximum number of iterations. In Algorithm 1, the AMIS pseudocode is displayed.

---

**Algorithm 1: Artificial Multiple Intelligence System (AMIS)**

---

input: Population Size (NP), Problem Size (D), Mutation Rate (F), Recombination Rate (R), Number of Intelligence Boxes (NIB)
output: Best_Vector_Solution
***begin***
    *Population = Initialize set of WPs*
    *IBPop = Initialize InformationIB(NIB)*
    *encode Population to WP*
        ***while*** *the stopping criterion is not met **do***
            ***for*** *i=1: NP*
                *$V_{rand1}$, $V_{rand2}$, $Vr_{and3}$ = Select_Random_Vector (WP)*
                ***for*** *j = 1:D // Loop for the mutation operator*
                *$V_y$ [j]= $V_{rand1}$ [j]+ F ($V_{rand2}$ [j]+ $V_{rand3}$ [j])*
                ***end for loop****//end mutation operator*
                ***for*** *j = 1:D //Loop for recombination operation*
                ***if*** *(randj [0,1) <R) then*
                *u [j]= $V_i$ [j]*
                ***else***
                *u [j]= $V_y$ [j]*
                 ***end for loop****//end recombination operation*
                *// selected Intelligence box by RouletteWheelSelection*
                *selected_IB = RouletteWheelSelection(IBPop)*
                ***if*** *(selected_IB = 1) then*
                *new_u = AIM (u)*
                ***else if*** *(selected_IB = 2)*
                *new_u = PIM (u)*
                ***else if*** *(selected_IB = 3)*
                *new_u = DIM (u)*
                ***else if*** *(selected_IB = 4)*
                *new_u = BIM (u)*
                ***else if*** *(selected_IB = 5)*
                *new_u = RT (u)*
                ***else if*** *(selected_IB = 6)*
                *new_u = IT (u)*
                ***else if*** *(selected_IB = 7)*
                *new_u = RT-AIM (u)*
                ***else if*** *(selected_IB = 8)*
                *new_u = SF (u)*

| Algorithm 1: *Cont.* |
| :--- |

> *else if* *(selected_IB = 9)*
>     *new_u = RESTART (u)*
> *if* *(CostFunction(new_u)* $\leq$ *CostFunction(V$_i$)) then*
>     *V$_i$ = new_u*
> *// Loop for updating the intelligence box's heuristics data*
> *for* *j = 1: NIB*
>     *interpreting WP to discover the real problems solution*
>     *Gather Pareto Front Data and Calculate TOPSIS*
> *end* *For Loop//end update heuristics information*
>     *end* *for Loop*
>       *end*
>     *return* *Best_Vector_Solution*
> *end*

### 4.3. The Compared Methods

In order to compare with the AMIS, we use two heuristics: the genetic algorithm (GA) and the differential evolution algorithm. The GA algorithm is based on the work of Mitchell [51], and the DE approach used in this study is based on the work of Sethanan and Pitakaso [52]. The Pareto front method is used by GA and DE, as has been previously mentioned.

### 5. Framework and Results of the Computation

The mathematical model is coded in Lingo V.16, and the AMIS, DE and GA are coded in C++ and run on a PC with an Intel ® Core TM i5-2467M CPU running at 1.6 GHz. To compare the performance of the proposed methods, 16 randomly generated datasets and 1 case study are used. The computational results are divided into two groups: Section 5.1 the revealed effectiveness of the AMIS and Section 5.2 the AMIS's behaviors in dealing with the proposed problem.

### 5.1. Demonstrate the Effectiveness of AMIS in Solving Random and Real-World Problems

The mathematical model presented in Section 3 is inapplicable to solving large-scale problems. Only a small test instances can be executed when comparing the AMIS, DE and GA to the solution obtained from Lingo v.16 used to solve the proposed problem. However, random data sets that mimic the real-world problem are required to validate the effectiveness of the AMIS. A total of 15 data sets are generated at random using the range of parameters provided by the real-world problem. Table 5 shows the specifics of the 15 data sets and the case study, and Table 6 shows the range of parameters used in the experiments.

**Table 5.** Information about the random data set and the case study.

| Instance Name | Number of Tourist Groups (Group) | Number of Tourists (Person) | Number of Attractions/ Destinations | Planning Period |
| :---: | :---: | :---: | :---: | :---: |
| W-1 | 5 | 97 | 72 | 7 |
| W-2 | 5 | 133 | 72 | 10 |
| W-3 | 5 | 184 | 84 | 12 |
| W-4 | 8 | 319 | 84 | 12 |
| W-5 | 8 | 510 | 84 | 12 |
| W-6 | 8 | 582 | 84 | 16 |
| W-7 | 15 | 914 | 84 | 16 |
| W-8 | 15 | 941 | 96 | 16 |

**Table 5.** *Cont.*

| Instance Name | Number of Tourist Groups (Group) | Number of Tourists (Person) | Number of Attractions/ Destinations | Planning Period |
|---|---|---|---|---|
| W-9 | 15 | 1091 | 96 | 16 |
| W-10 | 30 | 2190 | 96 | 16 |
| W-11 | 35 | 2584 | 96 | 20 |
| W-12 | 40 | 2962 | 96 | 20 |
| W-13 | 40 | 3084 | 121 | 20 |
| W-14 | 40 | 3157 | 121 | 30 |
| W-15 | 50 | 3955 | 121 | 30 |
| C-1 | 50 | 4143 | 137 | 30 |

**Table 6.** The parameter range of the case study used to generate the data sets.

| Parameters | Range | Parameters | Range |
|---|---|---|---|
| LN | 500,000 | $p_i$ (Persons) | [80, 460] |
| $u_{ni}$ (Points) | [4, 10] | $o_i$ (Points) | [4, 10] |
| $v_{njt}$ (Persons) | [0, 10] | $c_i$ (Baht) | [100, 500] |
| $w_{ik}$ | [0, 1] | $\varnothing_s$ (Baht) | [400, 1000] |

The first experiment compares the suggested approach (AMIS) with the Lingo v.16 solution. Lingo v.16's computational time is set to 240 h or 14,400 min in case it cannot find the optimal solution, and the computational time is recorded in Table 6 when Lingo v.16 can find the optimal solution. The stopping criteria for AMIS uses the computational time, which is set to 30 min for all test instances. Using the Technique for the Order of Preference by Similarity to the Ideal Solution (TOPSIS), a promising solution is discovered. Initially, TOPSIS was introduced by Hwang and Yoon [50]. TOPSIS first builds a normal decision matrix and then converts the dimensions of different attributes into a non-dimensional attribute using (26) to (32).

$$r_{lv} = \frac{x_{lv}}{\sqrt{\sum_{l=1}^{L} (X_{lv})^2}} \tag{26}$$

$$U_{lv} = w_v r_{lv} \tag{27}$$

$$U_v^* = \{ \max_L U_{lv} \ \ if \ \ v \in V \ ; \ \min_L U_{lv} \ \ if \ \ v \in V^* \} \tag{28}$$

$$U_v' = \{ \min_L U_{lv} \ \ if \ \ v \in V \ ; \ \max_L U_{lv} \ \ if \ \ v \in V' \} \tag{29}$$

$$S_l^* = \sqrt{\sum_{v=1}^{V} (U_v^* - U_{lv})^2} \tag{30}$$

$$S_l' = \sqrt{\sum_{v=1}^{V} (U_v' - U_{lv})^2} \tag{31}$$

$$C_l^* = \frac{S_l'}{S_l^* + S_l'} \tag{32}$$

where $l$ is the number of points in the Pareto front, $V^*$ is a set of positive objective functions, $V'$ is a set of negative objective functions and $x_{lv}$ is the value of the objective function of point l objective v. The weight of an objective function is the specified parameter, or $w_v$. The best solutions for the positive and negative cases are U* ($U^* = \{ U_1^*, U_2^*, \ldots, U_n^* \}$) and U′

($U' = \{U'_1, U'_2, \ldots, U'_n\}$), respectively. The separation measurements from the positive and negative ideal solutions, $S^*_l$ and $S'_l$, are used to determine how far apart each alternative is from the ideal answer. The promising solution is the set of parameters with a $C^*_l$ value that is closest to 1. Table 7 presents the values of two objectives using a different set of $w_1$ and $w_2$ of the 16 test instances, including the case study.

**Table 7.** Lingo v.16 and AMIS computational results when $w_1$ is the weight of the satisfaction score, using different pairs of weights for $w_1$ and $w_2$.

| | Lingo v.16 | | | | | | | AMIS | | | | | | |
| | $w_1 = 0.5, w_2 = 0.5$ | | $w_1 = 0.1, w_2 = 0.9$ | | $w_1 = 0.9, w_2 = 0.1$ | | Com. Time (Minutes) | $w_1 = 0.5, w_2 = 0.5$ | | $w_1 = 0.1, w_2 = 0.9$ | | $w_1 = 0.9, w_2 = 0.1$ | | Com. Time (Minutes) |
| | Satisfaction (Score) | Total Cost (Baht) | Satisfaction (Score) | Total Cost (Baht) | Satisfaction (Score) | Total Cost (Baht) | | Satisfaction (Score) | Total Cost (Baht) | Satisfaction (Score) | Total Cost (Baht) | Satisfaction (Score) | Total Cost (Baht) | |
|---|---|---|---|---|---|---|---|---|---|---|---|---|---|---|
| W-1 * | 6630 | 70,120 | 6567 | 65,490 | 6740 | 125,282 | 339.3 | 6630 | 70,120 | 6537 | 65,490 | 6719 | 125,282 | 10.7 |
| W-2 * | 10,162 | 117,722 | 10,041 | 112,588 | 10,232 | 143,272 | 347.2 | 10,162 | 117,722 | 10,041 | 115,583 | 10,232 | 143,272 | 20.8 |
| W-3 * | 27,997 | 405,824 | 27,417 | 314,071 | 27,997 | 405,824 | 523.8 | 27,997 | 405,824 | 27,417 | 314,071 | 27,997 | 405,824 | 25.8 |
| W-4 * | 45,982 | 757,865 | 45,578 | 695,853 | 46,930 | 779,765 | 1410 | 44,235 | 757,865 | 45,578 | 722,991 | 46,930 | 782,182 | 27.5 |
| W-5 | 82,399 | 1,684,090 | 74,040 | 865,194 | 83,080 | 1,838,974 | 14,400 | 83,214 | 1,590,197 | 74,040 | 816,983 | 83,080 | 1,738,453 | 30.0 |
| W-6 | 102,388 | 2,160,524 | 91,089 | 1,043,798 | 102,900 | 2,297,151 | 14,400 | 108,309 | 2,070,869 | 96,933 | 981,769 | 108,173 | 2,223,621 | 30.0 |
| W-7 | 200,578 | 3,810,915 | 175,886 | 2,268,883 | 201,028 | 3,857,902 | 14,400 | 215,479 | 3,694,219 | 192,482 | 2,187,206 | 213,386 | 3,670,236 | 30.0 |
| W-8 | 204,244 | 3,677,865 | 183,049 | 2,099,596 | 208,091 | 4,902,770 | 14,400 | 221,560 | 3,559,632 | 192,941 | 2,030,103 | 225,331 | 4,621,271 | 30.0 |
| W-9 | 234,452 | 5,037,064 | 196,084 | 3,267,995 | 234,518 | 5,226,302 | 14,400 | 248,674 | 4,744,578 | 210,172 | 3,122,413 | 247,004 | 5,062,970 | 30.0 |
| W-10 | 312,062 | 6,528,241 | 262,297 | 4,224,412 | 312,233 | 6,637,461 | 14,400 | 328,910 | 6,308,218 | 283,337 | 3,984,404 | 342,317 | 6,405,986 | 30.0 |
| W-11 | 472,811 | 9,807,354 | 401,569 | 6,602,266 | 473,845 | 10,006,565 | 14,400 | 513,967 | 9,266,786 | 421,954 | 6,306,535 | 513,688 | 9,584,266 | 30.0 |
| W-12 | 550,270 | 11,380,035 | 472,981 | 8,016,179 | 550,880 | 11,588,320 | 14,400 | 591,410 | 10,761,718 | 519,552 | 7,634,403 | 601,189 | 11,207,614 | 30.0 |
| W-13 | 527,260 | 10,736,330 | 470,415 | 6,675,515 | 531,362 | 12,052,420 | 14,400 | 564,491 | 10,372,998 | 514,450 | 6,381,725 | 565,991 | 11,478,608 | 30.0 |
| W-14 | 671,727.5 | 13,439,990 | 594,357.6 | 7,183,665 | 680,654 | 16,378,220 | 14,400 | 737,957 | 12,859,685 | 635,782 | 6,953,417 | 725,454 | 15,513,864 | 30.0 |
| W-15 | 804,707.1 | 15,607,910 | 720,143.3 | 8,887,651 | 818,796.8 | 20,538,430 | 14,400 | 850,843 | 15,032,954 | 756,633 | 8,401,172 | 881,846 | 19,756,930 | 30.0 |
| C-1 | 861,358.3 | 16,976,090 | 772,848.6 | 8,694,322 | 864,695.3 | 18,319,670 | 14,400 | 944,089 | 16,443,455 | 820,222 | 8,413,863 | 921,988 | 17,265,483 | 30.0 |

Note: * Optimal solution found, and others are found for objective bound after 240 h.

Table 7 demonstrates that the AMIS provides a better answer than the best result from Lingo version 16 after 240 h of calculation. The case study is included in the other test examples, and the AMIS can find 70.1 percent optimal answers in the first four problem instances. The AMIS's answer is around 3.83 percent more effective than Lingo v.16's. Lingo v.16 takes an average of 10,963.77 min to locate the solution, whereas AMIS finds it in an average of 27.8 min. We can draw the conclusion that the AMIS generates better solutions and requires less computational time than Lingo v.16. The Wilcoxon signed-rank test was used to determine whether there is a significant difference between Lingo V.16 and the AMIS. With a $p$-value of 0.00094, we discovered that the AMIS considerably outperforms Lingo V.16 in terms of discovering better solutions.

Table 8 compares the proposed approaches (AMIS) for solving M-ADes-SFWFPT to Lingo V.16, DE and GA, and it shows how they differ in terms of percentage. Utilizing a formula, the percentage difference is determined (33).

$$\%diff = \frac{\sum_{v=1}^{V} \left| f^v_{AMIS} - f^v_r \right|}{\sum_{v=1}^{V} f^v_p} \times 100\% \tag{33}$$

where $f^v_p$ is the objective function of objective $v$ of the suggested approach (AMIS), $v$ runs from 1 to $V$, and $f^v_r$ is the best objective function found in Lingo V.16 after 240 h of calculation. The GA and DE are produced when r = 1, 2, and 3, respectively.

According to Table 8, Lingo v.16, GA and DE offer less effective solutions than the AMIS. In the GA and DE, the average AMIS departure from Lingo v.16 is 3.83, 7.50 and 8.17 percent, respectively. GA and DE use the same 30 min computation time as the AMIS for all test instances. To assess whether there is a significant difference between the suggested methodology and the current method, the results of Table 8 show that the AMIS greatly raises the caliber of the solutions generated by all of the previously mentioned methodologies. According to the results of the Wilcoxon signed-rank test, which examines the data in Table 8, the AMIS significantly improves the quality of the solutions compared

to GA and DE, with a *p* value of 0.00005. This is due to the fact that the AMIS uses more effective improvement strategies than other methods. However, the AMIS is used in the case study, and the solution and outcome are shown in Section 5.2.

**Table 8.** % difference between AMIS and other approaches.

| Method | Lingo v.16 (240 h) | | | GA | | | DE | | |
|---|---|---|---|---|---|---|---|---|---|
| Instances | $w_1 = 0.5$, $w_2 = 0.5$ | $w_1 = 0.1$, $w_2 = 0.9$ | $w_1 = 0.9$, $w_2 = 0.1$ | $w_1 = 0.5$, $w_2 = 0.5$ | $w_1 = 0.1$, $w_2 = 0.9$ | $w_1 = 0.9$, $w_2 = 0.1$ | $w_1 = 0.5$, $w_2 = 0.5$ | $w_1 = 0.1$, $w_2 = 0.9$ | $w_1 = 0.9$, $w_2 = 0.1$ |
| W-1 * | 0.00 | 0.23 | 0.16 | 4.88 | 5.16 | 6.73 | 1.73 | 10.76 | 9.39 |
| W-2 * | 0.00 | 1.33 | 0.00 | 3.39 | 4.30 | 6.67 | 2.63 | 5.54 | 3.20 |
| W-3 * | 0.00 | 0.00 | 0.66 | 4.68 | 7.39 | 5.10 | 4.07 | 3.13 | 9.63 |
| W-4 * | 4.45 | 1.95 | 0.16 | 5.26 | 2.83 | 8.65 | 4.55 | 2.29 | 2.87 |
| W-5 | 3.13 | 6.72 | 3.32 | 2.25 | 7.46 | 12.47 | 5.45 | 6.39 | 5.96 |
| W-6 | 6.49 | 3.74 | 5.24 | 8.50 | 7.04 | 9.04 | 8.60 | 9.30 | 7.56 |
| W-7 | 4.97 | 10.50 | 0.58 | 6.36 | 6.47 | 2.81 | 9.86 | 11.38 | 11.31 |
| W-8 | 3.61 | 1.99 | 3.32 | 10.19 | 7.36 | 7.65 | 5.83 | 5.96 | 10.31 |
| W-9 | 6.18 | 3.71 | 4.68 | 5.92 | 8.49 | 6.25 | 10.43 | 10.84 | 11.74 |
| W-10 | 8.52 | 3.92 | 3.66 | 9.50 | 7.25 | 7.24 | 8.66 | 9.54 | 9.19 |
| W-11 | 1.13 | 2.55 | 6.92 | 9.81 | 8.56 | 13.71 | 12.01 | 7.18 | 7.13 |
| W-12 | 7.95 | 8.69 | 5.19 | 8.37 | 6.88 | 12.51 | 13.25 | 12.52 | 9.21 |
| W-13 | 4.87 | 3.09 | 7.01 | 7.83 | 4.93 | 10.21 | 13.44 | 5.81 | 10.94 |
| W-14 | 3.31 | 7.41 | 2.54 | 4.75 | 9.70 | 10.15 | 6.79 | 7.62 | 14.31 |
| W-15 | 2.76 | 4.32 | 1.60 | 8.04 | 6.94 | 11.07 | 7.82 | 9.29 | 8.12 |
| C-1 | 4.28 | 8.77 | 8.30 | 10.36 | 7.35 | 11.77 | 9.64 | 12.22 | 6.85 |
| Mean | 3.85 | 4.31 | 3.33 | 6.88 | 6.76 | 8.88 | 7.80 | 8.11 | 8.61 |
| Max | 8.52 | 10.50 | 8.30 | 10.36 | 9.70 | 13.71 | 13.44 | 12.52 | 14.31 |
| Min | 0.00 | 0.00 | 0.00 | 2.25 | 2.83 | 2.81 | 1.73 | 2.29 | 2.87 |
| Grand Mean (%) | 3.83 | | | 7.50 | | | 8.17 | | |

Note: * Optimal solution found, and others are found for objective bound after 240 h.

Comparing the existing well-known heuristics such as GA and DE to the new proposed AMIS in a multi-objective manner is the next experiment that we will perform. When evaluating how well GA and M-VaNSAS perform in solving the proposed problem, the average ratio of the Pareto-optimal solution (ARP) is employed as a measure of their effectiveness. Let N1, N2, ... Nk represent the number of iterations used in experiment k. The number of Pareto-optimal solutions found in the kth experiment is represented by n1, n2, ... , nk, where K is the total number of experiments. As a result, Equation (34) is used to determine the ARP. For the purpose of executing E.34, the number of GA, DE and AMIS tests iterations varies from 250 to 1500 iterations, and the number of Pareto points is counted before being converted to the value of Equation (34). It has been discovered that the AMIS, DE and GA have average Pareto points of 2611, 2194 and 2123, respectively, and their corresponding ARPs are 0.50, 0.42, and 0.41.

$$ARP = \frac{\frac{n_1}{N_1} + \frac{n_1}{N_2} + \ldots + \frac{n_k}{N_k}}{K} \tag{34}$$

Inferring that the AMIS is superior to the GA and DE in terms of discovering more solutions, we can see from the computational results in Table 8 that it discovers 22.99 and

19.00 percent more Pareto points. The Pareto fronts of the AMIS, GA and DE are compared in Figure 4.

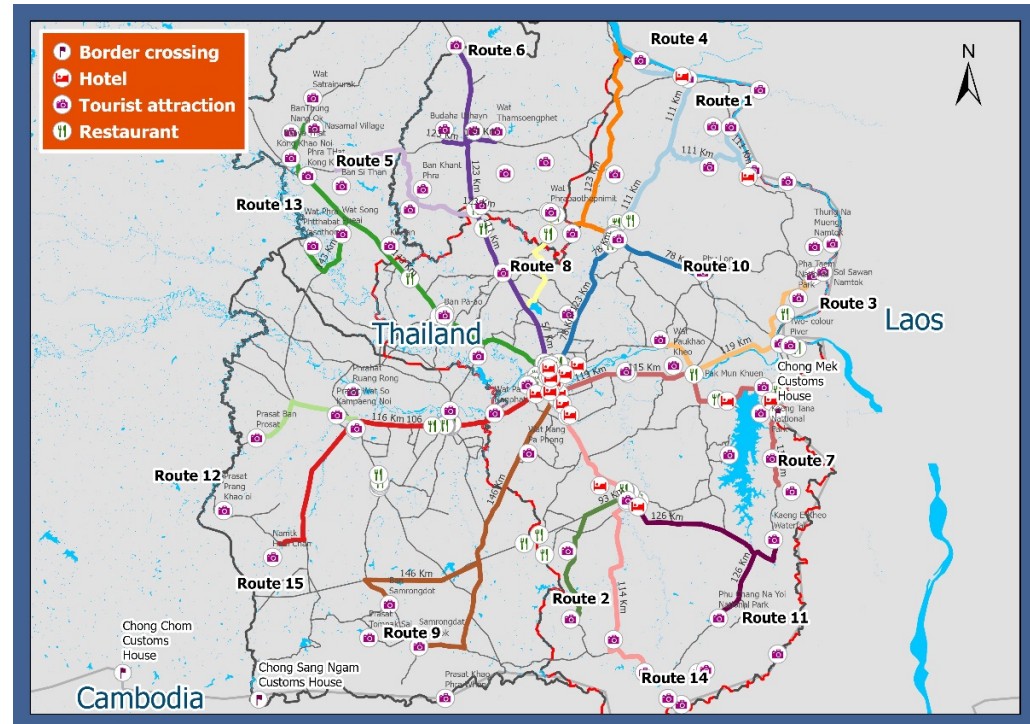

**Figure 4.** Pareto front of AMIS, GA and DE.

Figure 4 demonstrates that the GA and DE's Pareto fronts have wider gaps between the Pareto points than the AMIS. As a result of the data we gathered, the GA and DE discover 2123 and 2194 Pareto points after 1500 iterations, whereas AMIS discovers 2611 points after the same number of iterations. The likelihood of GA and DE having a bigger gap between each Pareto point is higher since their ARPs are, respectively, 21.93 percent and 19.25 percent less than that of AMIS, as shown in Figure 4.

### 5.2. Case Study Results

This case study is given the number C-1. It entails tourism planning for Ubon Ratchathani's province of wellness tourism for 50 groups, totaling 4143 tourists over a 30-day period in December 2022. Four categories—stress reduction (SR), sickness prevention (SP), mental health improvement (MI) and physical health improvement—are used to categorize the 99 wellness facilities and attractions (PI). Figure 5 depicts the distribution of Ubon Ratchathani Province's 99 attractions.

**Figure 5.** A total of 99 wellness tourism attractions of Ubon Ratchathani.

The attractions shown in Figure 5 can be categorized to be at least one type of attractions (PI, MI, SR and SP).

There are 50 distinct tour groups that enter the wellness tourism sector at various times. There are 30 periods total during the planning period. The case study is resolved using the AMIS, with 1000 iterations serving as the ending point and 99 constituting the number of Work Packages (WP) for each iteration. A total of 100 iterations constitute the sub-iteration iteration (from the preliminary test). Figure 6 displays the results of the attraction assignment for the group tour.

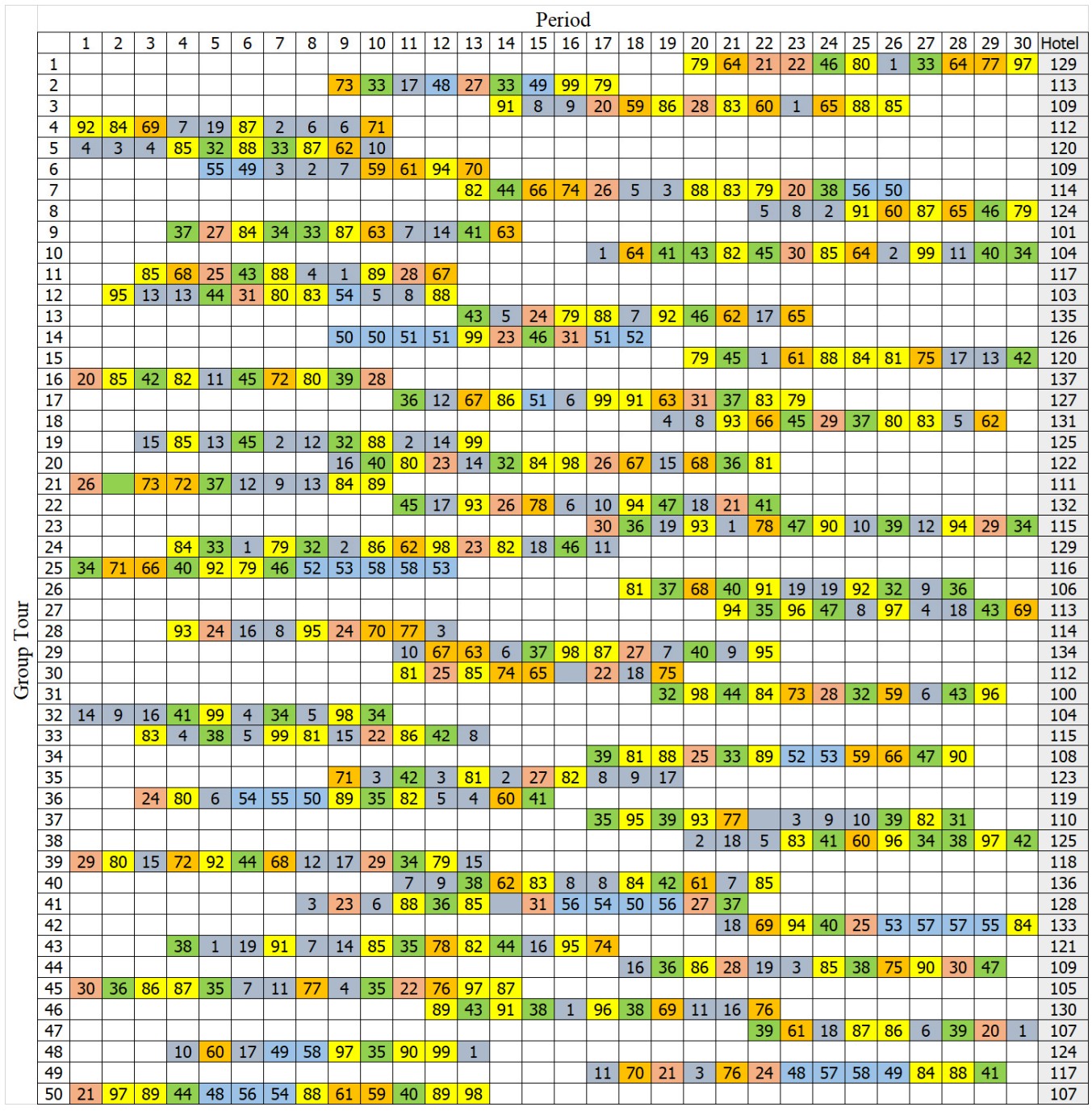

**Figure 6.** Assignment of the case study results (Gantt chart of the group tour).

A schedule of the tourist groups participating in the wellness tourism system in the target area is shown in Figure 6 (case study). The attractions assigned to the particular tourist group over the specified time period are indicated by the number, whereas the types of attractions are indicated by color codes in the Gantt chart. The outcome of the hotel assignment is shown in the last column of Figure 6. The Gantt chart for the hotels designated to serve the tourist group is shown in Figure 7. The hotel's use during the teaching period is indicated by the number inside the Gantt chart.

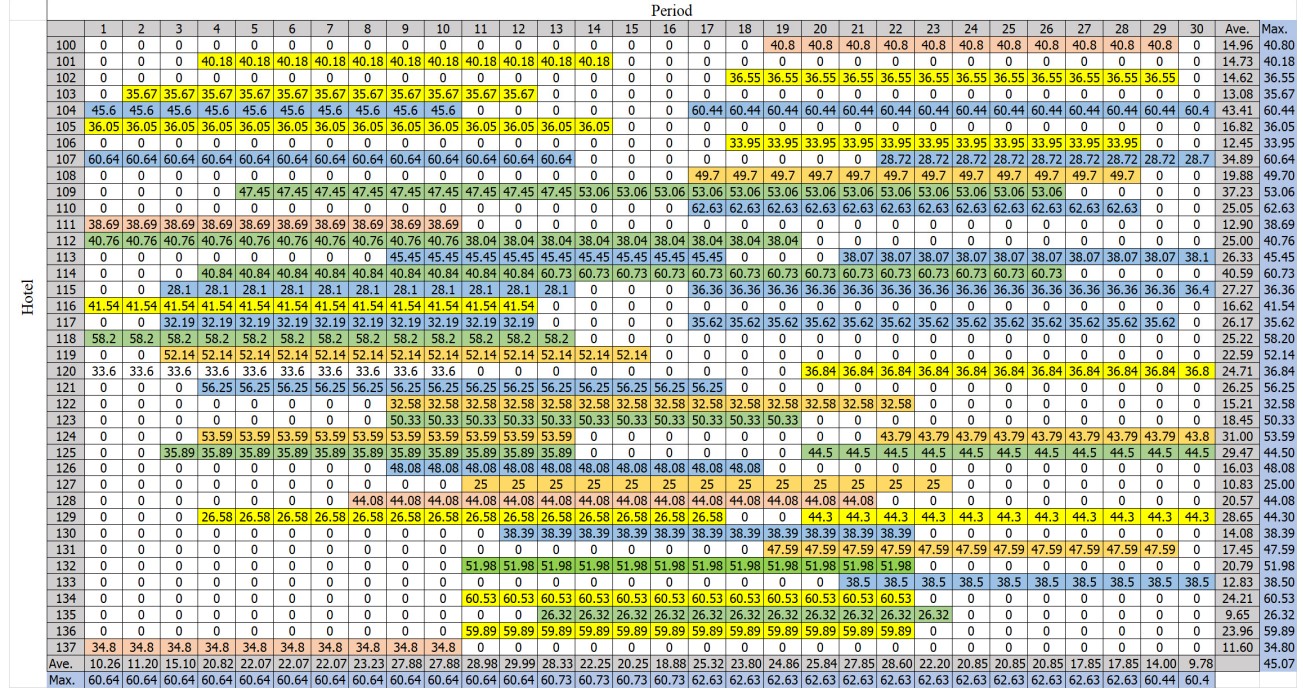

**Figure 7.** Gantt chart of the hotels and their utilization.

From Figure 7, the average utilization of the hotels is 21.72% of their full capacity. This means that they can serve more tourists if they have more tourists entering the system. Using the information in Figure 7, the times that the hotels are filled with guests regardless of the number of guests they are hosting can be determined. Occupancy rates at the hotel range from 33.37 to 83.33 percent, with an average of 50.09 percent. Therefore, there are more than 49% of free times periods available for new visitors. However, the experiment is run to show that, as more people travel, the system is used more frequently and earns more money.

Tables 9 and 10 present the time and percentage of each group of attractions (Table 9) and type of attraction (Table 10).

**Table 9.** Percentage of time that groups of attractions and destinations are visited.

|  | SPA (SPA) | Restaurant (RES) | Outdoor Recreation (OUT) | Beauty Service (BEA) | Temple (TEM) | Must-See Attraction (MU) |
|---|---|---|---|---|---|---|
| Visit time | 142 | 52 | 104 | 42 | 78 | 149 |
| Percentage | 25.04 | 9.17 | 18.34 | 7.41 | 13.76 | 26.28 |

**Table 10.** Percentage of time that types of attractions and destinations are visited.

|  | PI | MI | SR | SP |
|---|---|---|---|---|
| Number of visit time | 152 | 140 | 250 | 112 |
| Percentage | 23.24 | 21.41 | 38.23 | 17.13 |

According to the findings in Tables 9 and 10, the "Must-See Attraction" is the case study's most well-liked attraction and destination, having been visited 149 times or 26.28 percent of the time by group tours during a 30-day period. Spas are the second most popular location, and beauty salons are the least popular (BEA). When we examine the different sorts of attractions and destinations, we can discover that stress-relieving (SR) attractions are the most frequented sites by tourists compared to the other three tourist types.

Figure 8 demonstrates the three group of tourists in the case study. Each group is assigned destinations based on their preferences. For example, group 1 focuses on outdoor recreation and spas in the city, group 2 focuses on staying and using facilities in the boutique hotel, and group 3 focuses on the must-see attractions.

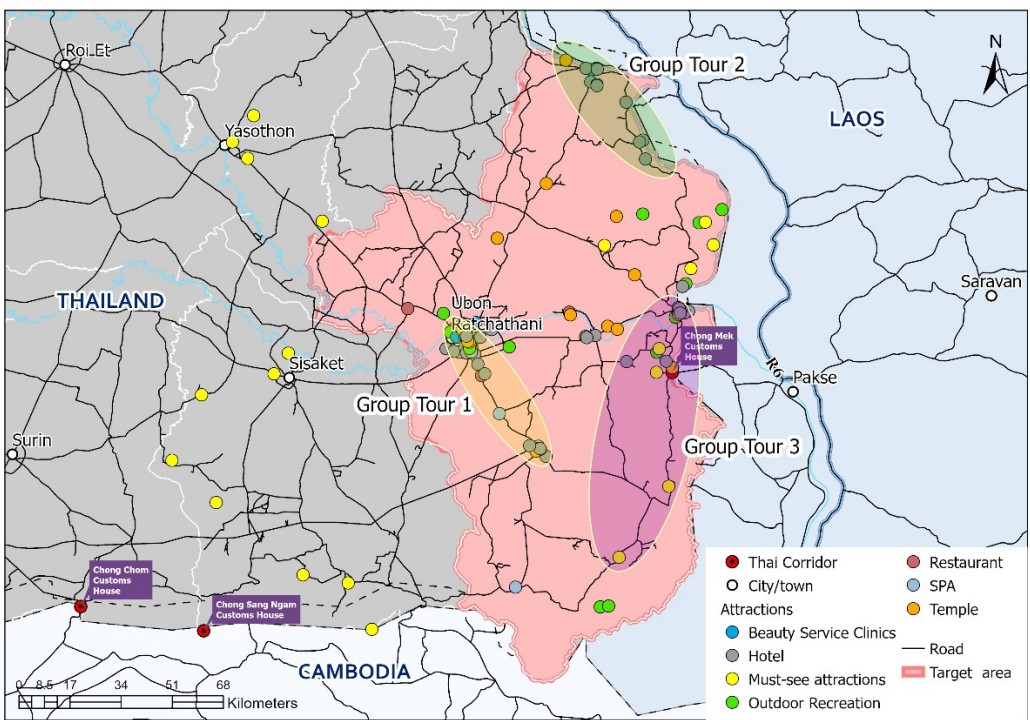

**Figure 8.** Demonstration of wellness tourist groups and attractions in the case study.

As previously noted, the hotel is utilized during all 30 time periods at a rate of about 50.09 percent. Hotels should, therefore, be able to accommodate more guests. We conduct the following experiment by increasing the number of tourists and the number of groups of tourists by percentages of 10%, 15%, 20%, 25%, 30%, 35, 40% and 50% from the existing real-world case problem in order to determine the overall profit made by the system when the number of tourists increases. Equation (1a) is adjusted to use Equation (35) as the objective function, and Equation (1b) is left unaltered from Section 2.

$$Max \ Z^3 = \sum_{n=1}^{J} \sum_{t=1}^{T} \sum_{j=1}^{j} \sum_{i=1}^{I} v_{njt} \Psi_i X_{ijt} \tag{35}$$

where $\Psi_i$ is the profit generated by attraction $i$ per tourist per time. The value of the profit generated by i is between 158 and 420 Baht per person per day. $w_1$ and $w_2$ are set to be 0.5 and 0.5, respectively. The simulation results are shown in Table 11.

The tourist revenue per visit generated by attraction $i$ is $\Psi_i$. The amount of the profit made per individual per day ranges from 158 to 420 Baht. A value of 0.5 is set for $w_1$, and 0.5 is set for $w_2$. Table 11 displays the results of the simulation.

**Table 11.** Profit and total satisfaction of the proposed methods using different levels of number of tourists.

| % Increase in Number of Tourist Groups/Number of Tourists | # of Tourist Groups | # of Tourists | Total Profit (Baht) | | | Total Satisfaction (Score) | | |
|---|---|---|---|---|---|---|---|---|
| | | | AMIS | GA | DE | AMIS | GA | DE |
| 0% | 50 | 4143 | 42,258,600 | 39,350,121 | 38,360,405 | 941,100 | 916,740 | 917,222 |
| 5% | 53 | 4351 | 45,353,375 | 41,910,145 | 39,706,733 | 987,999 | 926,999 | 926,349 |
| 10% | 55 | 4558 | 48,038,863 | 44,328,025 | 40,447,324 | 1,035,180 | 934,267 | 935,162 |
| 15% | 58 | 4765 | 50,345,098 | 48,360,307 | 47,653,714 | 1,082,067 | 978,306 | 979,885 |
| 20% | 60 | 4972 | 51,181,304 | 48,427,079 | 48,782,534 | 1,129,515 | 999,468 | 1,022,546 |
| 25% | 63 | 5179 | 56,348,527 | 52,514,727 | 51,611,057 | 1,176,273 | 1,074,458 | 1,072,772 |
| 30% | 65 | 5386 | 57,211,453 | 53,838,960 | 52,895,189 | 1,223,016 | 1,121,959 | 1,122,233 |
| 35% | 68 | 5594 | 60,367,640 | 55,978,474 | 54,964,127 | 1,270,530 | 1,169,431 | 1,146,359 |
| 40% | 70 | 5801 | 62,097,400 | 57,008,199 | 56,956,233 | 1,317,043 | 1,212,261 | 1,170,367 |
| 45% | 73 | 6008 | 64,283,979 | 58,552,392 | 57,859,578 | 1,364,364 | 1,259,638 | 1,212,925 |
| 50% | 75 | 6215 | 65,710,741 | 60,613,434 | 58,639,101 | 1,410,588 | 1,306,686 | 1,294,481 |
| | | | 54,836,089 | 50,989,260 | 50,716,000 | 1,176,152 | 1,081,838 | 1,072,755 |

According to Table 11, the AMIS, GA and DE see a rise in profit when the total number of tourists and tourist groups increases by a maximum of 50%. This indicates that, even with a 50% increase in tourists, the attractions still have ample capacity. It is clear that there is still sufficient demand for the hotel, destinations and attractions due to the information given in Table 11. Table 12 shows how a hotel's time and capacity change as the number of tourists traveling for wellness increases.

**Table 12.** Utilization of a hotel using different levels of tourists.

| % Increase in Number of Tourist Groups/Number of Tourists | # of Tourist Groups | # of Tourists | Utilization (Percentage) | | | Percent Occupied Servicing Periods (Percentage) | | |
|---|---|---|---|---|---|---|---|---|
| | | | AMIS | GA | DE | AMIS | GA | DE |
| 0% | 50 | 4143 | 21.72 | 19.23 | 18.58 | 50.09 | 45.41 | 43.94 |
| 5% | 53 | 4351 | 24.75 | 23.74 | 23.07 | 54.27 | 50.18 | 47.34 |
| 10% | 55 | 4558 | 28.15 | 23.89 | 26.55 | 56.55 | 50.69 | 51.75 |
| 15% | 58 | 4765 | 31.66 | 25.76 | 29.68 | 59.06 | 54.32 | 54.65 |
| 20% | 60 | 4972 | 34.87 | 30.56 | 33.37 | 60.30 | 57.90 | 55.17 |
| 25% | 63 | 5179 | 38.13 | 34.21 | 34.16 | 65.17 | 59.14 | 58.15 |
| 30% | 65 | 5386 | 41.83 | 36.69 | 37.58 | 65.61 | 64.09 | 59.19 |
| 35% | 68 | 5594 | 44.10 | 40.40 | 42.01 | 70.55 | 64.96 | 61.31 |
| 40% | 70 | 5801 | 48.11 | 43.06 | 43.41 | 73.61 | 65.12 | 64.49 |
| 45% | 73 | 6008 | 52.71 | 45.87 | 46.98 | 77.37 | 65.96 | 66.68 |
| 50% | 75 | 6215 | 53.76 | 48.00 | 51.12 | 81.90 | 66.46 | 69.07 |
| average | | | 38.16 | 33.76 | 35.14 | 64.95 | 58.57 | 57.43 |
| % different of 50% increase and 0% increase | | | 147.5 | 149.6 | 175.2 | 63.5 | 46.3 | 57.2 |

Table 12 demonstrates that, as more visitors enter the system, the proportion of the hotels that are utilized rises for all methods employed to address the issue. The AMIS, GA and DE enhance earnings by 147.5 percent, 149.6 percent and 175.2 percent, respectively, when tourism increases by 50 percent, and these three strategies raise the hotel's time utilization by 46.3 percent to 63.5 percent.

In comparison to other approaches, the AMIS can raise the overall profit from the GA and DE by 17.47 and 20.3 percent, and it can raise the preference score from the GA and DE by 17.96–23.47 percent, respectively. When we divide the percent change in the number

of tourists into two levels, (1): 0–25 percent and (2): 26–50 percent, the final report of this research is utilized to demonstrate if the trend of the total profit is altered. The outcome is displayed in Table 13.

**Table 13.** Slope of the change in profit and preference score when using different levels of number of tourists, changed from the current situation.

| Range | Profit | | Preferable Score | |
|---|---|---|---|---|
| | Average Profit | Slope | Average Profit | Slope |
| 0–25% | 48,920,961 | 0.33 | 1,058,689 | 0.25 |
| 26–50% | 61,934,243 | 0.15 | 1,317,108 | 0.15 |

Table 13 demonstrates that the initial 25% of the growth in visitor numbers has a steeper slope in both profit and preference score than the last 25% of the growth. The final 25% of the growing tourist population only has a slope of 0.15 and 0.15 for profit and preference score, and the first 25% has a slope of 0.33 for profit and 0.25 for preference score. This is due to the fact that there is no guarantee that, when the number of tourists rises, they are directed to attractions and destinations that create larger profits; occasionally, however, this is necessary due to the limitations of the attractions and destinations.

## 6. Discussion and Recommendations

The discussion and recommendation section is divided into two sub-sections, which are the academic implications and the business and social implications of this research.

### 6.1. Academic Implications

The proposed approach, the AMIS, outperforms the solution produced by the optimization software (Lingo V.16), which has a limited computing time (240 h), according to the experimental results displayed in Section 5.1. The AMIS finds a solution that is 3.83% better than the one found by the optimization tool using only 30 min of calculation time. Due to Lingo v.16's exact method, where the computing time increases exponentially as the size of the problem increases linearly, more time is required to find the best solution as the number of the test instances grows. As a result, Lingo v.16 takes longer than the proposed method, the AMIS, to identify the best solutions. Sethanan and Pitakaso [52], Santini [53], Tsakirakis et al. [54] and Bayliss et al. [55] give evidence to support the notion that heuristics take less processing time than optimization tools such as Lingo v.16.

In addition to outperforming the optimization software Lingo v.16, the suggested technique, the AMIS, also provides a superior solution compared to well-known existing heuristics, such as the genetic algorithm (GA) and differential evolution algorithm (DE). While using the same amount of computational time, the AMIS delivers a better solution than GA and DE by 7.50% and 8.17%, respectively. This is due to the AMIS's use of more effective improvement approaches, such as RT-AIM, Restart, Random Transit, Inter-Transit, PSO-Inspired Move, DE-Inspired Move and ACO-Inspired Move, which can enhance the quality of the existing solutions' solutions. These AMIS improvement techniques include enhancement techniques that may boost the system's capacity for both the intensification and diversification of searching. The AMIS is capable of successfully escaping the local optimal and can conduct a thorough search to identify the best solution in a certain search area. This conclusion is reinforced by Sangkaphet et al. [56] and Khonjun et al. [57], who also draw the conclusion that their approach considerably improves the quality of GA and DE solutions because it incorporates more efficient improvement techniques.

The AMIS outperforms the GA and DE in finding the better result while using the computational time, according to this study and the conclusions. The AMIS can be used by researchers to tackle a variety of combinatorial optimization problems, including the vehicle routing problem, the traveling salesman problem, the location–allocation dilemma, the production lot sizing problem and the facility location problem. The AMIS developed

in this study can also be used for other types of optimization problems, such as fine-tuning friction stir welding settings and identifying the ideal sets of parameters for heliostat design. This study also provides an approach for resolving the multi-objective decision problem, a topic that is now a hot one among research teams. The design of delivery routes for customer purchases to minimize travel expenses while maintaining high levels of customer satisfaction is an example of a multi-objective decision-making dilemma.

*6.2. Business and Social Implications*

The implications of this research for business and society are divided into two categories: (1) local SME tourism actors in the supply chain for wellness tourism; and (2) the travel agency, an actor in the chain that is responsible for creating the itinerary for wellness visitors.

The mathematical model that generates the combined SME wellness tourism business sectors (WTBS) to be the wellness tourism sectors that are efficient is provided in this study. WTBS generally do not draw wellness tourists because they are initially modest and unable to meet the needs of a particular group of visitors that want various forms of wellness tourism attractions. An increased ability to service clients and, naturally, the interests of all parties involved in the wellness tourism industry result from the establishment of more robust economic sectors that can fulfill tourist demand. According to [37], cooperation between and within enterprises can be a crucial component of the chain's success. This result is consistent with our study's finding that the cooperation of many business sectors in the same wellness tourism supply chain can boost the profit of the entire chain.

Within the confines of the test problem's small size, a mathematical model can handle the problem optimally. The quantity of tourists, tourist groups and tourist sites all influence the size of the problem case. The biggest size of the case study has 4143 tourists, which makes it inapplicable to solve to optimality by utilizing optimization software. Lingo v.16 can solve to optimality when the number of tourists is under 319 people.

In addition to being able to handle problems of any size, the AMIS also outperforms the best currently available heuristics such as GA and DE in terms of performance. The AMIS can increase the overall profit from the GA and DE by 17.47 and 20.3 percent, respectively, in contrast to these methods. Therefore, in order to expand their complete company chain, the commercial sectors can utilize the AMIS to identify the timetable of the tourist groups planning to visit. The proposed methodology is intended to solve similar problem types that try to determine a supply chain's client schedules. This approach can be used to address the issue of cooperating business sectors, such as supply chains for medical tourism, semi-conductors and agriculture. As an illustration, consider the collaboration of a hospital and a hotel, a hospital and a hospital, a hospital and a hotel with a transportation company, etc.

**7. Conclusions**

In order to identify potential wellness destinations and attractions in Thailand's Ubon Ratchathani Province, this research tries to choose the best combination of small and medium-sized destinations and attractions. Using Figure 3 as the selection framework, 46 wellness attractions/destinations are chosen based on the selection criteria. Based on the rating given by visitors/tourists who have been there, the quality of the attractions and locations is determined. In addition to attempting to minimize assignment costs, the selection process's primary goal is to increase the preference scores of the visitors who visit the chosen sites. The model that is the focus of this research is planned at the operational planning level. The attractions/destinations and the travel agency have both received a tentative schedule, however. Tourists that visit for wellness purposes arrive and depart at various times. Different consumer categories, including those of varying ages and genders, influence the tourist group's desire to travel for wellness. Different customer types favor different attractions in different ways. It is more challenging to choose attractions or places for each group of visitors because each customer group is composed of various tourist types.

The mathematical model of the selection of the locations and attractions is solved using Lingo version 16 (optimization software). Unfortunately, Lingo v.16 can only solve the problem for the first 4 test instances (because they small test instances), and it is unable to solve the remaining 11 instances, including the case study (because they are large test instances), to the best possible solution based on the type of computers and computational time that are available.

The AMIS has been created to address the suggested issue. The AMIS has five steps that it uses to find the optimum answer. Nine intelligence boxes are intended for use in order to raise the caliber of the existing solution. The AMIS is a type of metaheuristic; therefore, it cannot guarantee that the best solution is found, but it can guarantee that the right answer is found in an acceptable amount of time. The two existing heuristics that operate in a similar way to the AMIS are the genetic algorithm and the differential evolution algorithm. According to the calculation, the AMIS can improve solution quality by 3.83 to 8.17 percent on average across all test instances. As a result, we can say that the AMIS performs better than all other techniques at locating the optimal solution. When the number of visitors doubles, the AMIS, GA and DE can direct visitors to the attractions that make 29.76, 29.58 and 32.20 percent; 29.76 percent; and 32.20 percent, respectively, more income than they currently make while maintaining the same level of preference. The greatest increase in utilization from the existing scenario for the attractions/destinations is 265.4 percent. This implies that small and medium-sized businesses have a greatly increased probability of succeeding in the marketplace.

## 8. Limitations and Outlook

As previously mentioned, the model in this article is built on the framework shown in Figure 3. There are four different categories of wellness tourist attractions/destinations, as shown in Figure 3. If more intriguing locations are discovered by wellness travelers, the variety of wellness attractions and destinations may grow. One may classify education as another form of wellness tourism, for instance, because many courses in education, such those of yoga, cooking, English, music and dancing, can help people decompress and relieve stress. Food tourism is another form of wellness tourism that is not included in our framework. The variety of food included in wellness attractions/destinations, such as organic food restaurants, local food restaurants, healthy food restaurants, sweet food creperies and cooking schools, makes it possible to categorize food separately from other types.

The wellness tourism supply chain depends on collaboration between two business sectors in order for the various business sectors to work together. If the working conditions are unsuitable for both actors, this may lead to conflict between the sectors that have agreed to work together. To create a schedule that works well for tourists, industry sectors must standardize their working practices and exchange their information. The actors in the chain must all exchange information, including the costs and capacity needed to run their businesses, according to this extension of the model that has been provided. As a result, the relationships between the actors in the chain must be solid and based on mutual trust.

These are some ways that this study can be expanded upon: (1) because it may have an impact on the system's logistics costs, the routing of tourist transportation should be taken into account when visitors alter the site of their service; (2) the AMIS needs to be thoroughly investigated, including the reasons for its success in resolving logistics issues and other related fields, as well as the primary theory behind it.

**Author Contributions:** Conceptualization, writing—original draft preparation, formal analysis, R.P. and N.N.; Conceptualization, methodology, writing—review and editing S.D., K.C. and R.P.; project administration, R.P.; writing—review and editing, validation, S.K., T.S., W.S., G.J. and S.S.; writing—original draft preparation, R.P. and W.S.; software, N.N., S.K., T.S. and C.C. All authors have read and agreed to the published version of the manuscript.

**Funding:** This research was funded by Thailand Science Research and Innovation (TSRI), and National Science, Research and In-novation Fund (NSRF). Grant No. 2450857.

**Institutional Review Board Statement:** Not applicable.

**Informed Consent Statement:** Not applicable.

**Data Availability Statement:** Not applicable.

**Acknowledgments:** This work was supported by (i) the Faculty of Engineering, Ubon Ratchathani University (UBU), (ii) the Faculty of Industrial Technology, Ubon Ratchathani Rajabhat University, (iii) Thailand Science Research and Innovation (TSRI) and (iv) the National Science, Research and Innovation Fund (NSRF).

**Conflicts of Interest:** The authors declare no conflict of interest.

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
