# Peer review of "Solving the Optimal Selection of Wellness Tourist Attractions and Destinations in the GMS Using the AMIS Algorithm"

_computation, doi:10.3390/computation10090165_

Round 1

Reviewer 1 Report

The paper is well written and offers a lot of information which in the conclusion section should be translated in advices/recommendation for both academic and business - added value of the research should be highlighted. Next steps, problems faced and proposals for further imporvements should be summarized in the last part too. 

Author Response

Thank you for your valuable comments and suggestions. We tried to improve our article in every single point of your recommendations. Please see the attachment for responses.

Reviewer 2 Report

I appreciate the amount of work that the authors put into this article, but I must admit that it requires further changes and improvements in order to make it a useful scientific work.

1. The first remark is, unfortunately, the impression that the article is overloaded. I mean a lot of elements, tables, charts that are redundant. Please select the most important elements and describe the rest, possibly some tables and figures can be included in the appendix to the article (appendix). Too much information is not at all conducive to understanding the scientific argument. Currently, it seems like an abbreviation from a wider study.

2. The huge disadvantage of this work is the lack of discussion. A broad discussion (before the conclusions) should be developed, referring to the literature reports in the second chapter.

3. In the first chapter (Introduction) it is necessary to define precisely what is the purpose of this work, as it is not known at present. Only the next part of the article determines its practical usefulness. However, I remind you that this is a scientific article, so it is necessary to define what this work brings to science? At the same time, it is worth considering who this study is to serve, because what the authors presented suggests that their solution is to increase the use of the market potential. However, these assumptions are very doubtful, if only due to the fact that increasing the effectiveness of tourism potential may cause other problems and limitations, e.g. overtourism. So I ask again - who is it for? What are the expected scientific benefits? What are the limitations of the conducted analyzes?

Good luck in thoroughly improving your work.

Author Response

(The authors gave the same response as above.)

Round 2

Reviewer 2 Report

I believe that the article is now properly prepared, so I recommend publishing it.